# Programmably tiling rigidified DNA brick on gold nanoparticle as multi-functional shell for cancer-targeted delivery of siRNAs

Chang Xue 1, Shuyao Hu[1], Zhi-Hua Gao[2], Lei Wang[3], Meng-Xue Luo[1], Xin Yu[1], Bi-Fei Li[1], Zhifa Shen[2] & Zai-Sheng Wu 1✉

Small interfering RNA (siRNA) is an effective therapeutic to regulate the expression of target genes in vitro and in vivo. Constructing a siRNA delivery system with high serum stability, especially responsive to endogenous stimuli, remains technically challenging. Herein we develop anti-degradation Y-shaped backbone-rigidified triangular DNA bricks with sticky ends (sticky-YTDBs) and tile them onto a siRNA-packaged gold nanoparticle in a programmed fashion, forming a multi-functional three-dimensional (3D) DNA shell. After aptamers are arranged on the exterior surface, a biocompatible siRNA-encapsulated core/shell nanoparticle, siRNA/Ap-CS, is achieved. SiRNAs are internally encapsulated in a 3D DNA shell and are thus protected from enzymatic degradation by the outermost layer of YTDB. The siRNAs can be released by endogenous miRNA and execute gene silencing within tumor cells, causing cell apoptosis higher than Lipo3000/siRNA formulation. In vivo treatment shows that tumor growth is completely (100%) inhibited, demonstrating unique opportunities for next-generation anticancer-drug carriers for targeted cancer therapies.

[1] Cancer Metastasis Alert and Prevention Center, Fujian Provincial Key Laboratory of Cancer Metastasis Chemoprevention and Chemotherapy, Pharmaceutical Photocatalysis of State Key Laboratory of Photocatalysis on Energy and Environment, College of Chemistry, Fuzhou University, Fuzhou 350108, China. [2] Key Laboratory of Laboratory Medicine, Ministry of Education of China, Zhejiang Provincial Key Laboratory of Medicine Genetics, School of Laboratory Medicine and Life Sciences, Institute of Functional Nucleic Acids and Personalized Cancer Theranostics, Wenzhou Medical University, Wenzhou 325035, China. [3] Hunan Provincial Key Laboratory of Phytohormones and Growth Development, College of Bioscience and Biotechnology, Hunan Agricultural University, Changsha 410128, China. ✉email: wuzaisheng@163.com

RNA interference (RNAi) is an endogenous regulatory pathway capable of regulating about 30% human gene expression in the post-transcriptional level, which is triggered by small interfering RNA (siRNA) that can target and suppress the expression of its complementary gene mRNA with high specificity[1,2]. Since the pioneering work on RNA interference[3,4], siRNAs are considered to be potential gene-silencing therapeutics and are extensively explored for treating various diseases, including cancers[5,6]. However, targeted delivery of intact siRNAs to specific gene molecules in vivo is a technological bottleneck for the transition to clinical application owing to the susceptibility of siRNAs to serum nuclease degradation, poor cell permeability and lack of targeting ability[7,8]. For example, once a lipid-based delivery formulation was developed, the first siRNA drug, Onpattro (patisiran), was approved by the US FDA for the treatment of peripheral nerve disease (polyneuropathy)[9].

In the past decades, numerous siRNA delivery vehicles, such as polymers, liposomes, dendrimers, and inorganic particles, have been developed in order to improve cargo encapsulation efficiency, extend systemic circulation time and facilitate cellular uptake[10,11]. The researchers can impart tumor-targeting capabilities to these synthetic delivery nanoconstructs through functionalizing their surfaces with cancerous cell-specific binding ligands (e.g., aptamers[12]). There are some inherent limitations to the clinical application of RNAi therapy, for example, the susceptibility to the attack by endogenous nucleases[11,13,14]. Auxiliary exogenous components, mainly including strong cationic polymers and their derivatives, often also cause serious cytotoxicity to normal organs[15,16]. For RNAi therapy to advance toward clinical use, it is highly desirable to develop innovative delivery systems with tumor-targeting property, controllable siRNA encapsulation, long blood circulation time and natural biodegradability in vivo, as well as sufficient design flexibility.

Moreover, the inadequate release of encapsulated siRNAs insides target tumor cells compromises the RNAi efficacy, constituting an obstacle preventing the translation of siRNA-based therapeutic strategy to the clinic[17]. As a class of non-coding RNAs that regulate the gene expression in a sequence-specific fashion, microRNAs (miRNAs) play a vital role during tumorigenesis, metastasis, and progression and become biomarker candidates for the clinical diagnosis and prognosis of cancers as well as potential therapeutic targets[18,19]. Naturally, miRNA is one of the most promising endogenous stimuli for the controlled release of incorporated siRNAs[20,21]. Moreover, as a potential endogenous stimulus, miRNA exhibits an enhanced therapeutic activity superior to other stimuli such as pH, ATP, metal ion, enzyme and light because of its synergistic gene regulation therapy[22,23]. Nevertheless, unlike chemotherapeutic agents, it remains a technical challenge to develop efficient delivery systems for miRNA stimulus-responsive controlled release of encapsulated siRNAs[24], mainly due to the structural complexity designed for binding to small cytoplasmic RNAs and for resistance to enzymatic degradation.

Given their unique properties such as multiple functional capabilities, tunable sizes and high surface area-to-volume ratio, gold nanoparticles (AuNPs) have shown great potential in various applications including diagnostics, molecular imaging, and biomedicine[25,26]. Functional oligonucleotide-modified AuNPs can enter mammalian cells in high quantities without any transfection agent and exhibit desirable in vivo biodistribution and pharmacokinetic properties, indicating a promising nanoplatform for the delivery of siRNAs[27,28]. Recently, AuNP-based siRNA delivery nanosystems have been reported to induce effective gene silencing in cancerous cells, tissues and organs without apparent cellular toxicity[29]. Their immunogenicity is 25-fold lower than the lipid-based counterparts[30]. Moreover, AuNPs have been safely employed for the treatment of rheumatoid arthritis for over 60 years[31]. Unfortunately, because of entering almost all mammalian cells[27], AuNPs coated with oligonucleotides often fail to distinguish tumor cells from normal cells[32], hampering their use in therapeutic applications.

In this work, an AuNP-oligonucleotides core/shell nanostructure is developed for efficient delivery of siRNAs and controlled release within target cells, where the AuNP core is surrounded by the interior shell composed of siRNA-encapsulated standing-up DNA self-assembled layer and the outer coating consisting of lying-flat aptamer-incorporated Y-shaped backbone-rigidified triangular DNA bricks. In view of the structure and composition, the constructed siRNA delivery nanosystem is called siRNA-encapsulated and aptamer-incorporated core/shell (CS) nanoparticle (siRNA/Ap-CS) that exhibits high serum stability, tumor-specific targeting ability, long circulation lifetime and controlled siRNA release in endogenous miRNA-responsive fashion. The siRNA/Ap-CS efficiently knocks down the expression of *Plk1* mRNA and PLK1 protein and exhibits the higher ability to induce the apoptosis of tumor cells than commercial transfection reagent lipofectamine 3000. In vivo treatment of cancer-bearing mice demonstrates that siRNA/Ap-CS can completely inhibit the growth of malignant tumor, affording a competitive therapeutic avenue for human cancers.

## Results

**Constructing core/3D shell nanostructure and mechanism for siRNA delivery.** Figure 1 describes the construction of multifunctional core/shell nanovehicle on the basis of the combination of unique properties of AuNP with 3D DNA assemblies, accompanied by the molecular mechanism for tumor-targeted delivery of siRNAs and controllable release within target cancer cell. DNA-modified AuNP is firstly prepared via covalently attaching thiolated ADCs (anchoring DNA complementary to miRNA) onto AuNP though thiol-gold chemistry[33] (step I). Then, the elongated siRNAs are densely "loaded" onto AuNP surface by hybridization with the surface-confided ACDs, forming siRNA/ADC-AuNP (step II). Subsequently, the exterior of siRNA/ADC-AuNP is tiled with aptamer-incorporated Y-shaped backbone-rigidified triangular DNA bricks (Ap-YTDB) by hybridization with the terminal complementary fragments of ADCs, generating CS-type siRNA-encapsulated vehicles, siRNA/Ap-CS. The Ap-YTDBs can non-covalently cross-interact with each other via hybridization reaction because they are designed to have palindromic sticky-ends at every vertex, forming the compact protective coating with unconventional ligands, aptamers, orienting outward from siRNA/Ap-CS surface (step III). The siRNA/Ap-CS is capable of resisting nuclease degradation and remaining substantially intact within the cells for a sustained period of time long enough for RNAi therapeutics. When introduced into living organisms, such as mouse model, siRNA/Ap-CS nanostructures can reach specific tissues and organs by the blood circulation system and enter into target cancer cells by aptamer–receptor interaction without the need of transfection reagents (step IV). Because the ADC strands are design to preferentially hybridize with cellular miRNAs, the encapsulated siRNAs are released from core/shell nanoparticles into the nucleoplasm owing to strand displacement reactions (step V) and enter the RNAi pathway, knocking-down the expression of tumor-promoting genes.

The high drug-loading capacity and adequate stability against in vivo degradation, as two critical issues to be addressed for siRNA delivery for cancer therapy, remain exciting and formidable challenges especially for the nanovehicles assembled from

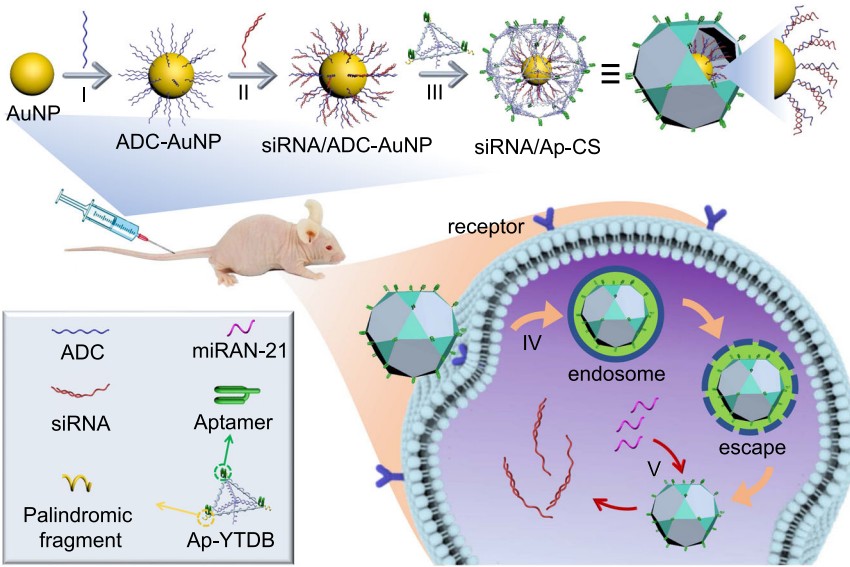

**Fig. 1 Preparation of gold nanoparticle/multi-functional 3D DNA self-assembled multilayer core/shell nanostructure (siRNA/Ap-CS) and its application for targeted siRNA delivery and controlled release within tumor cells. Step I**, functionalization of AuNP with thiol-modified anchoring DNA complementary to miRNA (ADC), forming ADC-AuNP; **Step II**, binding of elongated siRNA to AuNP-confined ADC, producing siRNA/ADC-AuNP; **Step III**, programmed positioning of aptamer (Ap)-integrated Y-shaped backbone-rigidified triangular DNA bricks (Ap-YTDB) onto siRNA/ADC-AuNP, leading to the formation of expected biocompatible core/shell nanostructure, siRNA/Ap-CS; **Step IV**, aptamer-mediated specific cell recognition and endocytosis; **Step V**, intracellular controlled release of encapsulated siRNAs by the strand displacement initiated by endogenous miRNAs.

nucleic acids that are susceptible to the degradation by endogenous nucleases. For the developed core/shell nanostructure, two striking structural features are the 3D structure of nucleic acid shell surrounding the inner AuNP core and the outermost protective coating consisting of sticky-YTDBs, making it suitable for the dense loading of siRNA and subsequent in vivo targeted delivery. Specifically, for the proposed nucleic acid shell, unlike the conventional 3D structure, for example, a three-dimensionally nanostructured pillars[32,34], besides the "standing up" (vertical) surface-confined ADC strands (the interlayer) on the surface of AuNP, there are lying-flat (horizontal) and cross-linked DNA bricks (forming the outer layer) over the vertical interior layer. Apparently, the hierarchical sandwich-type core/shell structure of AuNP core/vertical ADC interlayer/horizontal YTDB outer layer is assembled, generating the protective cavity in the interlayer for gene drug loading. For the protective outer coating (or also called outermost layer), a Y-shaped backbone is correctly installed into the center of basic structural unit, DNA triangle, substantially enhancing the structural rigidity of DNA bricks. According to previous study on the relationship between the structural features of DNA substrates and digestion ability of nucleases, the higher mechanical rigidity can more efficiently protect DNA compositions against enzymatic degradation[35]. Therefore, the developed DNA nanostructure is expected to become a promising carrier for the in vivo delivery of siRNA therapeutics to cancer tissues because of the ultra-protective outermost coating. Moreover, an array of fluorophores attached onto DNA components could open exciting avenues for different researches, such as monitoring the cellular internalization of siRNA/Ap-CS, observing the intracellular behaviors of CS nanostructure, assessing the expression level of cancer-related miRNA triggers and imaging the siRNA release.

**Characterization of siRNA/Ap-CS nanostructure.** Assembly of sticky-YTDB serving as the basic structural unit of 3D shell and its structural advantages are shown in Supplementary Note

("Assembly of sticky-YTDB and its structural advantages", Supplementary Figs. 1 and 2). In this section, several different DNA nanostructures are also assembled as controls.

To directly confirm the structural features of YTDB-a, the hierarchical assemblies from the three YTDBs (YTDB-a, YTDB-b, and TDB-c) were comparatively explored by visualizing by AFM. As shown in Fig. 2a, the cross-linked DNA assemblies on the basis of hexagonal tiles are observed from YTDB-a, while the assembled products from YTDB-b or YTDB-c are mostly separate (few aggregates are possibly produced during drying the sample under a stream of nitrogen) under identical conditions. The magnified AFM images show the hexagonal tiles whose assembly scheme is illustrated in the middle panel. Each hexamer is composed of six YTDB-a units via sticky end-based cross-linking. DNA hexamers can further assemble into more complicated nanostructures because of the hybridization between the residual out-pointing palindromic sticky ends at the vertexes. The formation of YTDB-a-based hexagonal tiles and their further assembly are consistent with a fact that triangular DNA nanoscaffolds with sticky ends are prone to form hexagons[36] and thus they were used as the basic units to prepare the advanced hexagonal arrays[37]. Because of the mechanical stability, the sticky DNA triangles and the corresponding hexagon tile-based structures possess unique structural advantages as building scaffolds to harbor multifunctional moieties for different purposes, such as fluorescent bioassay and therapeutic intervention[38].

Then, siRNA/Ap-CS nanostructure was constructed as described in Methods, and its stepwise assembly was characterized by dynamic light scattering (DLS) where intermediate formulations serve as controls. As shown in Fig. 2b, the hydrodynamic diameter of several nanoparticles, including AuNP, ADC-AuNP, siRNA/ADC-AuNP, and siRNA/Ap-CS, increases from 35 nm, to 71 nm, to 73 nm and to 128 nm, demonstrating that the hydrodynamic thicknesses of ADC, ADC/siRNA, and ADC/siRNA/Ap-YTDB are 18 (36/2) nm, 19 (38/2) nm and 46.5 (93/2) nm, respectively. The structural analysis of nucleic acid shells and the theoretically-calculated thinness of

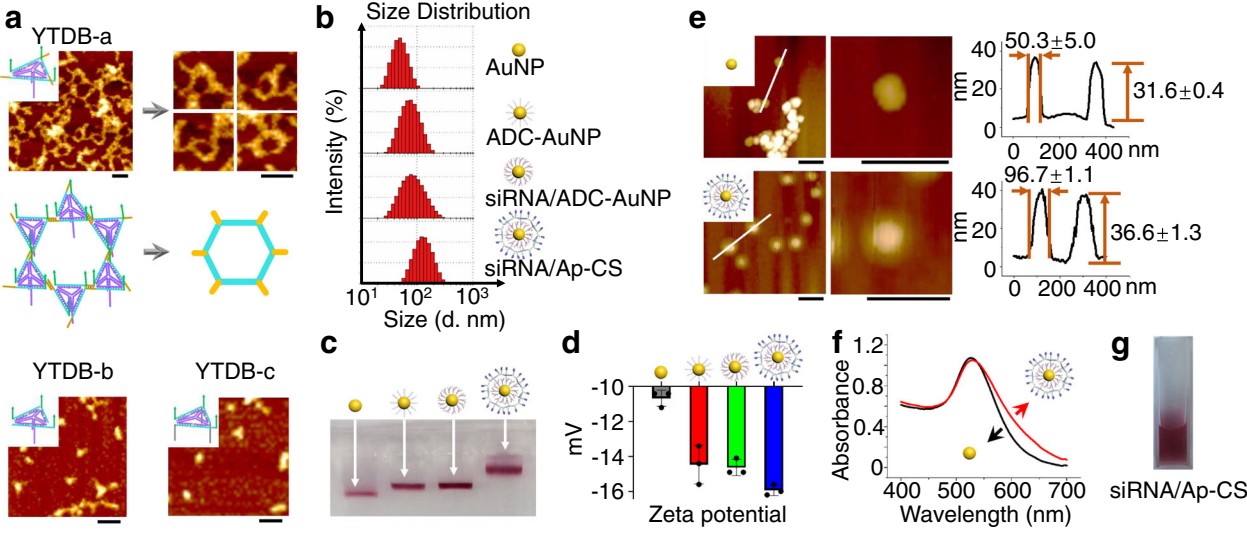

**Fig. 2 Characterization of siRNA/Ap-CS Nanostructure. a** AFM images of three Y-shaped backbone-rigidified triangular DNA brick (YTDB-a, YTDB-b, and YTDB-c). Inset: magnified AFM images of YTDB-a. Scare bar is 50 nm. **b** Dynamic light scattering (DLS) analysis and **c** agarose gel electrophoresis to confirm the stepwise assembly of siRNA/Ap-CS. The red bands indicate the naked or DNA functionalized AuNPs. These experiments were conducted three times independently with similar results. **d** Zeta potential analysis of four nanostructures in panel c mentioned by DLS. The error bar represents the standard deviation (SD). The measured data are expressed as the means ± SD (*n* = 3). **e** AFM images of naked-AuNPs and siRNA/Ap-CS. Middle panel: expanded view of a typical CS nanoparticle. Scare bar is 150 nm. Right panel: cross-section profiles of naked-AuNPs and siRNA/Ap-CS taken along the white line in the left AFM images. **f** UV−vis absorption spectrum of naked-AuNPs (black line) and siRNA/Ap-CS (red line). **g** Photographed image of siRNA/Ap-CS solution.

each layer are shown in Supplementary Fig. 4c. Assuming 0.34 nm/base pair for the helical pitch and 2 nm for the diameter of double-stranded (ds) DNA, the lengths of ADC, ADC/siRNA and ADC/siRNA/Ap-YTDB should be 17 nm, 17 nm and 42.4 nm, respectively. For the surface-confined nucleic acids (ADC), introduction of siRNAs leads to the rigid double-stranded structure and the more crowded geometries although not changing the length of nucleotide strands. Clearly, the apparent thickness of DNA shells on the three formulations obtained from DLS is slightly larger than the theoretically-calculated values, which is in accordance with literature observation[39]. The increasing trend of nanoparticle diameter versus stepwise assembly is consistent with the experimental results from agarose gel analysis. As shown in Fig. 2c, the electrophoretic mobility decreases in the order of AuNP>ADC-AuNP>siRNA/ADC-AuNP»siRNA/Ap-CS (red bands marked by white arrows represent naked or DNA functionalized AuNPs). Figure 2d shows that zeta potential of the four nanostructures sequentially decreases, indicating the increasing amount of nucleic acids assembled onto AuNP surface. The AFM analysis was also performed to estimate the size of siRNA/Ap-CS nanoparticles. As shown in Fig. 2e, compared to naked AuNP, there is a dense and relatively uniform corona surrounding a bright inner core (middle panel). The corona thinness is about 23 nm estimated from the difference in the diameter [(96.7−50.3) nm/2] between AuNP and siRNA/Ap-CS (right panel), which is smaller than the DLS value ([128−35] nm/2 = 46.5 nm) and theoretically-calculated value of nucleic acid shell (42.4 nm, Supplementary Fig. 4). The height of naked AuNP obtained is ~31 nm slightly smaller than DLS value (35 nm). That AFM values are lower than DLS data is reasonable, because the AFM measurements were conducted in dry form where the surface-confined biomolecule layer could shrink, while DLS measurements were performed in an aqueous medium where there is the hydration layer on AuNP surface[40]. For the AFM measurement, that the AuNP width is larger than its height is because of the well-known tip-broadening effect of AFM[41]. In addition, the red-shift of the UV/Vis

spectrum of core/shell nanoparticles is observed compared to naked AuNPs (Fig. 2f) because the assembly of protective outer coating induces the change in the dielectric constant of surrounding environment of AuNP[42], which is consistent with literature reports[18,43]. These experimental results indicate the successful assembly of siRNA/Ap-CS. Moreover, the siRNA/Ap-CS can be stably dispersed in an aqueous solution as demonstrated in Fig. 2g.

**YTDB-based protection of encapsulated siRNAs from enzymatic degradation.** Nuclease resistance of YTDB was confirmed as shown in Supplementary Fig. 3, while its assembly efficiency on nanoparticle surface was explored via quantitative fluorescence measurement in Supplementary Fig. 4. More supporting information is presented in Supplementary Note ("Nuclease resistance of YTDB and its assembly on nanoparticle surface"). Afterwards, the capability of YTDB-based shield to protect the siRNAs encapsulated in core/shell nanoparticles against nuclease degradation was evaluated. Firstly, the residual siRNAs after FBS treatment was quantified by fluorescence spectroscopy according to the procedure as shown in Fig. 3a. More information is described in Methods, while the experimental results are shown in Fig. 3b. One can see that the residual siRNAs in samples a to e are 99%, 92%, 72%, 37%, and 22%, respectively. We believe that the serum stability of siRNAs is directly related to the basic structure of protective outermost coating rather than to the YTDB number. It seems reasonable. For the cross-linked YTDB-a coating, because its structural units have no tendency to swing, it is not easy for the nucleases in FBS to attack. However, there is no the cross-interaction between lying-flat YTDB-b units or between standing-sideways YTDB-c units. Therefore, the nucleases in FBS readily degrade the YTDB-b layer and further the inner nucleic acids, including the encapsulated siRNAs. Especially, YTDB-c is most easily degraded because its free side or vertex points outward[35] due to the electrostatic repulsion from the inner negatively-charged nucleotides. More importantly, the percentage of residual siRNAs encapsulated in the YTDB-a-based core/shell

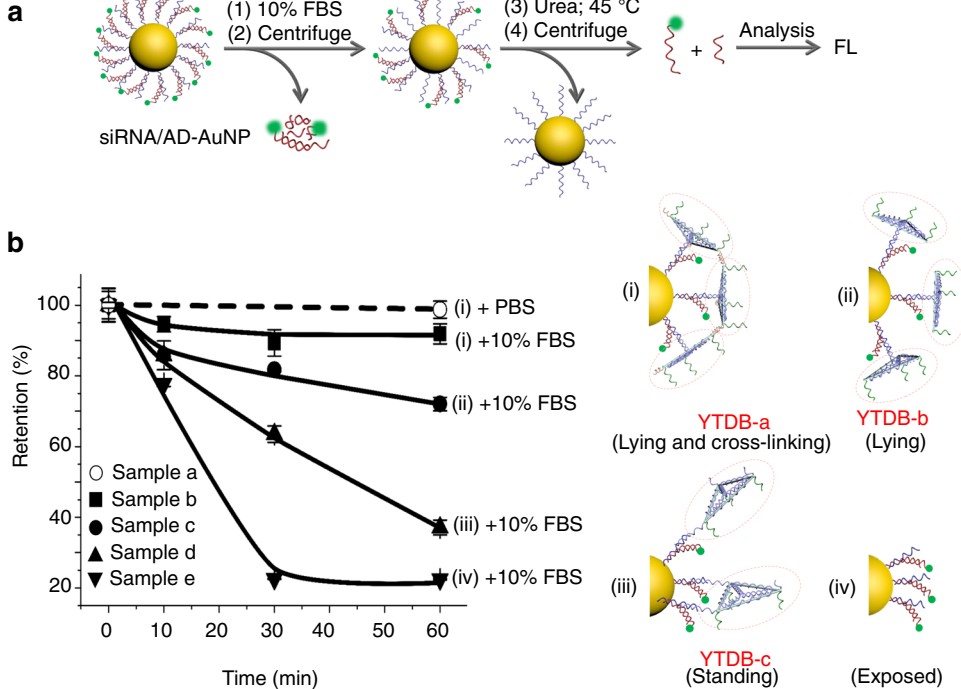

**Fig. 3 The fluorescently measured data in homogenous solution to demonstrate the FBS stability of siRNA/Ap-CS formulation, where siRNA was FAM-modified siLuc. a** Schematic illustration of the method for evaluation of the AuNP-confined siRNA stability. **b** Quantitative evaluation of the retention (%) of surface-confined siRNAs after exposure of four different siRNA-encapsulated formulations (i–iv) to FBS, where the retention efficiency of nontreated formulation corresponding to each group is defined as 100%. Samples a-e represent the solutions of Nanoparticle-i + PBS, Nanoparticle-i + 10% FBS, Nanoparticle-ii +10% FBS, Nanoparticle-iii +10% FBS, and Nanoparticle-iv +10% FBS, respectively. Nanoparticle-i is the siRNA/Ap-CS with the compact protective coating assembled from lying/cross-linking YTDB-a; Nanoparticle-ii and Nanoparticle-iii are the same as Nanoparticle-i but substituting YTDB-a with lying YTDB-b and standing YTDB-c, respectively. Nanoparticle-iv indicates siRNA/ADC-AuNP without the protective coating. More information can be seen in the section of "Retention (%) assay of siRNA on AuNPs after treatment with 10% FBS" in "Methods". The error bar represents the standard deviation (SD). The measured data are expressed as the means ± SD ($n = 3$).

nanoparticles is larger than those under the double protection offered by spherical nucleic acids (SNAs) and chemical modification (e.g., 2′-O-methyl, 2′-OMe)[44]. To provide the direct comparison, we evaluated the FBS stability of siRNAs encapsulated in the formulation with double protection, where the siRNAs were modified with 2′-OMe moieties. As shown in Supplementary Fig. 5, the FBS stability of siRNA/Ap-CS indeed is higher than the counterpart formulation with the SNA/2′-OMe-based double protection and cross-linked CS designed without YTDB.

To directly "visualize" the resistance of siRNA/Ap-CS formulation against FBS degradation, the dPAGE analysis was carried out and the results are presented in Supplementary Fig. 6. Clearly, the naked siRNA can be degraded within a few minutes (sample v). After attached onto AuNP, the serum stability of siRNA is improved to 10 min that is consistent with a literature study[44]. This is because the serum nuclease-catalyzed hydrolysis of surface-confined oligonucleotides can be inhibited by AuNPs, which has been studied by Chad A. Mirkin group[45]. After tiling the YTDB-c on the surface of AuNP, siRNAs could exist in FBS over 30 min. The higher serum stability of encapsulated siRNAs was achieved in samples i and ii. Moreover, the amount of residual siRNAs in sample i is substantially larger than sample ii. Further supporting evidence for the degradation resistance of siRNA/Ap-CS formulation is shown in Supplementary Fig. 7.

The measured data mentioned above demonstrate that tiling YTDB-a units onto siRNA-encapsulated nanoparticles endows the delivery vehicle with excellent capability to protect siRNAs from the nonspecific or specific enzymatic degradation. Since the nuclease-catalyzed hydrolysis of oligonucleotides conjugated onto

AuNP readily occurs[44] and chemical modification of siRNAs, such as at the 5′-terminus of antisense strand, potentially compromises the silencing activity[46], the protective YTDB-a-based siRNA/Ap-CS formulation with high siRNA loading capability without any chemical modification to siRNAs holds tremendous promise for RNAi-based cancer therapies.

**Targeted delivery, controlled release and gene silencing activity**. In vivo biodistribution and pharmacokinetics of siRNA/Ap-CS formulation were investigated, and the data show that the proposed 3D DNA shell-based nanovehicle exhibits the enhanced in vivo stability and desirable tumor targeting ("In vivo biodistribution and pharmacokinetics", Supplementary Figs. 8, 9, 10 and Fig. 4 in Supplementary Note). The stimulus-responsive release of siRNA can be realized because the ADC is capable of preferentially hybridizing with endogenous miRNA-21 ("Targeted Delivery, Controlled Release and Gene Silencing Activity", Supplementary Figs. 11 and 12, in Supplementary Note). After aptamer (AS1411) was installed onto the outermost protective layer of 3D DNA shell, the resulting nanovehicles can easily enter target HeLa cells by active binding to specific surface receptors ("Targeted Delivery, Controlled Release and Gene Silencing Activity", Supplementary Figs. 13 and 14 in Supplementary Note). As such, the desirable gene silencing activity of siRNA/Ap-CS is expected to be afforded, which was confirmed by the ability to suppress the luciferase expression ("Targeted Delivery, Controlled Release and Gene Silencing Activity" and Supplementary Fig. 15 in Supplementary Note). The expression level of luciferase was determined by luminescence measurement in luminometer

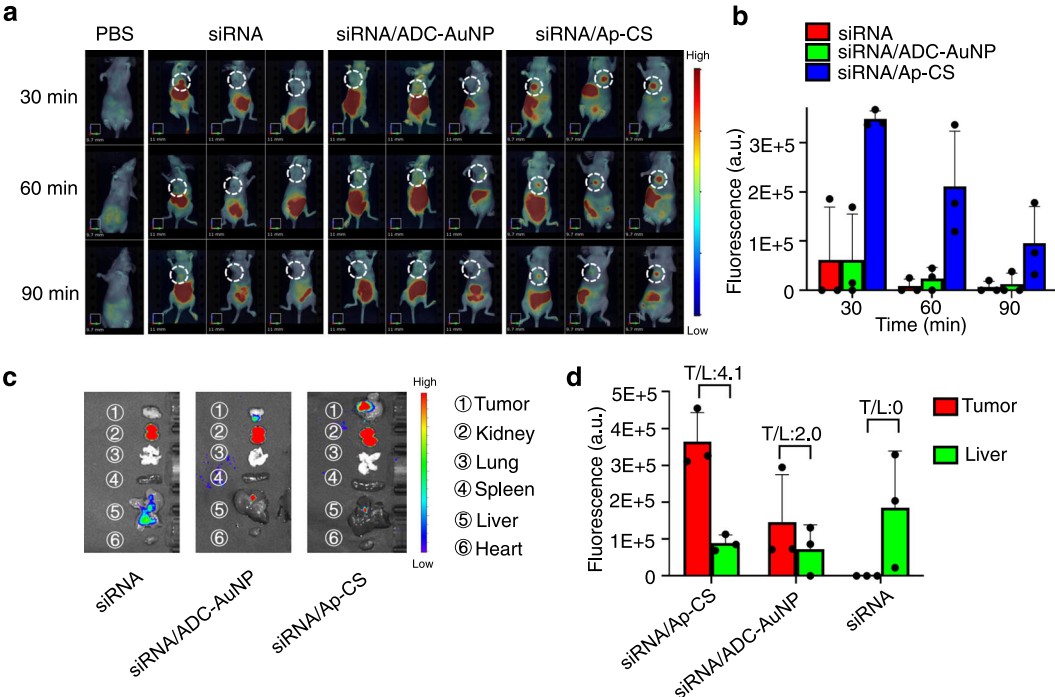

**Fig. 4 In vivo biodistribution of siRNA/Ap-CS formulation. a** Time-dependent in vivo fluorescence imaging to explore the whole-body biodistribution kinetics of siRNA/Ap-CS and its tumor localization. A549 tumor-bearing BALB/c nude mice were used, and the samples, including PBS, siRNA, siRNA/ADC-AuNP, and siRNA/Ap-CS, were administrated to the mice via tail vein injection. **b** The fluorescence intensity of tumor sites in the white circles in panel a. The error bar represents the standard deviation (SD). The measured data are expressed as the means ± SD (n = 3). **c** Fluorescence images of the organs harvested at 90-min post injection of siRNA/Ap-CS, accompanied by the quantitative assessment of fluorescence intensity from liver and tumor (**d**) where the fluorescence ratio of tumor-to-liver (T/L) is also presented. The PS-ADC (*Plk1* sense partly complementary to ADC) and PA-Cy5 (Cy5-labeled *Plk1* antisense) were used for the preparation of siPlk1 duplex. The error bar represents the standard deviation (SD). The measured data are expressed as the means ± SD (n = 3).

(Tecan Infinite 200 pro, Switzerland), and if the expression level of luciferase in HeLa cells treated with PBS is defined as 100%, only 27% luciferase is expressed after being treated with siRNA/Ap-CS. Namely, about 73% target genes is silenced, which is comparable with the commercial transfection reagent.

**Targeted delivery of siPlk1 to silence PLK1 expression in cancer cells**. The polo-like kinase-1 (PLK1) is an important regulator of the mitotic progression in mammalian cells[47,48], and its overexpression contributes to oncogenic transformation[49,50]. Silencing PLK1 expression via siRNAs can inhibit the cell cycle progression and initiates the apoptosis of cancer cells, holding significant promise for cancer therapy[51,52]. Thus, we constructed siPlk1/Ap-CS via formulating PLK1-specific siRNA, siPlk1, into core/shell nanoparticle and evaluated its potential for use in neutralizing PLK1 expression in cancerous cells. To directly observe the cellular internalization of siPlk1/Ap-CS nanoparticles, the cells treated with siPlk1 (modified with Cy5, red)-loaded formulations were visualized via fluorescence imaging by laser scanning confocal microscopy, accompanied by AuNP-based bright field imaging. As shown in Fig. 5a, a strong red fluorescence signal is detected in fluorescence image (Cy5) and a considerable number of AuNPs, some of which aggregate, are observed in bright field image (BF-image), demonstrating the successful cellular uptake of siPlk1/Ap-CS and intracellular release of formulated Cy5-labeled siPlk1. The specific release of Cy5-labeled siPlk1 from the nano-formulation is verified by fluorescence measurement as shown in Supplementary Fig. 12d. Unlike siPlk1/Ap-CS, Fig. 5b shows that, when un-siPlk1/Ap-CS was instead employed, the substantially compromised fluorescence signal is detected. This is because, even if cellular

internalization occurred, the un-siPlk1 was not efficiently released from the nano-formulation and thus Cy5 fluorescence was not restored (Supplementary Fig. 12e). Of note, almost as many AuNPs are seen in Fig. 5a as those in Fig. 5b, which is also confirmed by inductively coupled mass spectrometry (ICP-MS) (Supplementary Fig. 16). Figure 5c shows that, without aptamer on the outermost coating, almost no fluorescence signal is detected and very few AuNPs can be seen, implying only a very small number of siPlk1/CS nanoparticles can be internalized by cells and significantly inhibits the nanoparticle-mediated non-targeting cellular internalization. Figure 5d represents the experimental results similar to Fig. 5c, demonstrating that aptamer cannot promote the cellular uptake process of siPlk1/Ap-CS. This is because L02 cells, normal human hepatocytes, do not overexpress the corresponding cell surface receptors. These experimental results demonstrate that siRNA/Ap-CS formulation possesses the more desirable cellular permeability that is based on aptamer-mediated cancer cell-specific recognition and thus is able to execute the targeted siRNA delivery to tumor cells compared with the conventional nucleic acid-modified AuNPs similar to SNA[45,53–55]. The SNA is the well-known AuNP core/nucleotide shell nanoparticles that have be extensively studied as siRNA delivery system for the gene therapy of various cancers, including cutaneous tumors, glioblastoma, diabetes, and skin inflammation because of its low toxicity, low immunogenicity and persistent gene knockdown[27,56]. But, while significantly superior to their linear counterparts, SNAs still suffer from two inherent limitations: the extensive long-term in vivo susceptibility to nuclease degradation and the lack of targeting specificity[27,32]. The developed siRNA/Ap-CS formulation, capable of overcoming the two critical bottlenecks in systemic in vivo delivery of siRNAs to

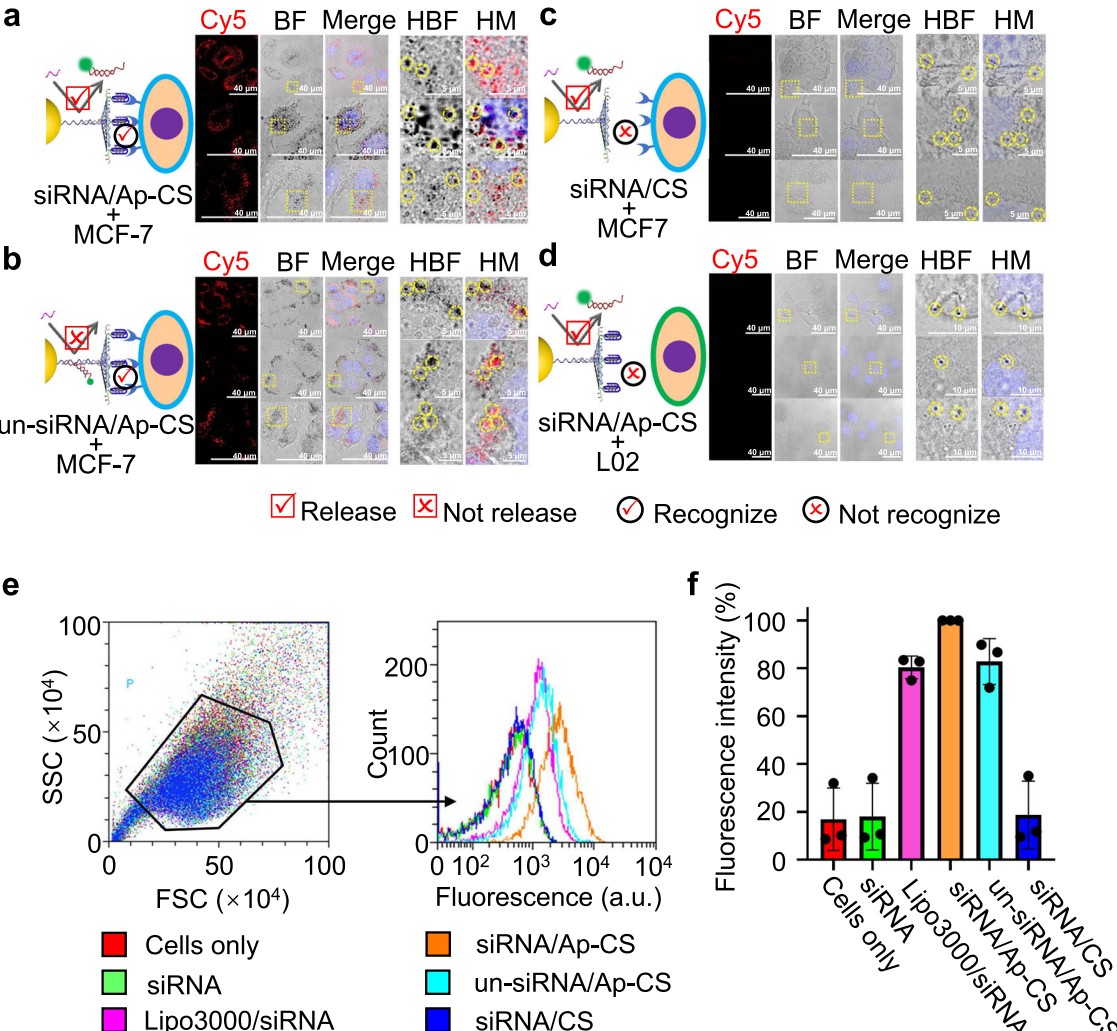

**Fig. 5 Colocalization assay of siRNA (red fluorescence) and AuNPs (black dot) within target cancer cells.** MCF-7 cells were separately incubated with releasable siPlk1/Ap-CS (**a**), unreleasable siPlk1 (un-siPlk1)/Ap-CS (**b**) and releasable siPlk1/CS without aptamer (**c**) for 4 h. **d** is the same as **a** but L02 cells were instead used. HM in the right half part is the high-resolution image of the area in yellow dotted box indicated in the section of Merge, while HBF is the high-resolution image boxed in bright field (BF). AuNPs in HM and HBF are highlighted by yellow dotted circles (**a–d**). The cell imaging experiments in **a–d** were conducted three times independently with similar results. **e** Flow cytometry analysis of MCF-7 cells treated with various formulations for 4 h. **f** The quantitative fluorescence intensity of each sample in **e**. The error bar represents the standard deviation (SD). The measured data are expressed as the means ± SD ($n = 3$).

tumor sites, provides an opportunity for the practical application of gene therapy in tumor diagnosis and clinical therapy.

Because the flow cytometry can offer statistic fluorescence data for a large population of cells and remove the unwanted variation encountered by some measurement techniques only involving a small amount of cells, including fluorescence image and bright field image by laser scanning confocal microscope, siPlk1/Ap-CS-incubated cells were analyzed by flow cytometry. As shown in Fig. 5e, f, we failed to detect the fluorescence signal for siPlk1 alone (green line) because siRNA duplex cannot enter cells; after transfected by lipofectamine 3000 (Lipo3000), the fluorescence signal is detected (red line), indicating the efficient internalization of siPlk1 into the cells; incubation with siRNA/Ap-CS induces a desirable fluorescence signal (light blue line) substantially higher than that upon lipo3000-based transfection, demonstrating the higher cellular uptake efficiency; for the un-siPlk1/Ap-CS, the treated cells only show a moderate optical signal (purple line). This is because un-siPlk1 cannot be displaced by intracellular miRNA, and its fluorescence is substantially quenched by AuNP. Nevertheless, the fluorescence intensity is still slightly higher than

that of Lipo3000-based transfection; without the aptamers hybridized on the core/shell nanostructure, almost no fluorescence signal is observed (yellow line). These measured data are consistent with the fluorescence images and bright field images mentioned above. A scramble siRNA was also used instead of siPlk1 to perform these experiments under identical conditions and similar results were obtained as shown in Supplementary Fig. 17. Moreover, the internalized siRNA/Ap-CS can successfully escape from endosome/lysosome into the cytosol (Supplementary Fig. 18), naturally contributing to the efficient gene silencing.

**In vitro validation of PLK1-targeted gene silencing and cell apoptosis.** After confirming the efficient internalization of siPlk1/Ap-CS into target cells, the gene silencing efficacy and therapeutic potency for cancers were investigated at the cellular level. For this purpose, the MCF-7 cells stably expressing *Plk1* were incubated with siPlk1-incorporated formulations for 48 h (the optimization of incubation time is shown in Supplementary Fig. 19) and then the real-time polymerase chain reaction (RT-PCR) experiments were conducted to evaluate the *Plk1* expression at the mRNA level. As

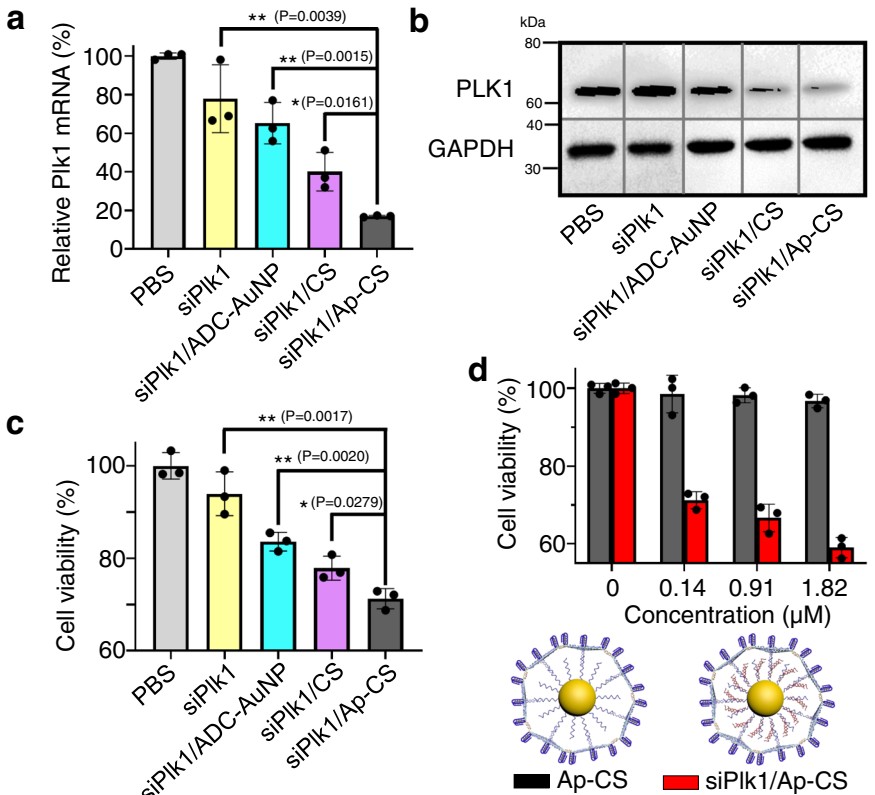

**Fig. 6 Capability of siRNA/Ap-CS to knocked down *Plk1* mRNA and PLK1 protein expression within MCF-7 cells.** The siRNA-targeting *Plk1* is abbreviated as siPlk1. After being treated with siPlk1 in different formulations, *Plk1* mRNA levels were measured by qPCR (**a**), PLK1 protein expression was analyzed by western blotting ($n = 3$) (**b**) and cell viability was test by CCK-8 kit (**c**). The western blot analysis was performed three times independently with similar results. **d** Cell viability of MCF-7 cells treated with different amounts of siRNA/Ap-CS. The concentration of siRNA in siRNA/Ap-CS ranges from 0 to 1.82 µM, where the value of 0 indicates that Ap-CS formulation without siRNA was used under identical conditions. The error bar in **a**, **c** and **d** represents the standard deviation (SD), and the measured data are expressed as the means ± SD ($n = 3$). *$P < 0.05$, **$P < 0.01$, two-tailed unpaired *t* test.

shown Fig. 6a, the trend in the endogenous expression of *Plk1* mRNA, PBS (100%) > siRNA (78%) > siRNA/ADC-AuNP (65%) > siRNA/CS (40%) > siRNA/Ap-CS (only 17%), is observed, demonstrating with certainty the critical roles of YTDB-based protective coating and the outside surface-arranged aptamers in the siRNA-mediated gene silencing. The expression of *Plk1* mRNA is fully in line with the PLK1 protein level determined by the western blot analysis (Fig. 6b) where one can see that PLK1 protein expression is almost completely suppressed by siPlk1/Ap-CS formulation.

To directly confirm the therapeutic potential of siPlk1/Ap-CS, the proliferation of MCF-7 cells after exposure to siPlk1 formulations for 48 h was checked using a CCK-8 assay. We noticed that, compared with other siRNA-loaded formulations, siRNA/Ap-CS can significantly inhibit the proliferation of MCF-7 cells (Fig. 6c), and the cell viability is reduced by siRNA/Ap-CS in a dose-dependent manner (Fig. 6d), implying a living cell-penetrating nanoplatform for efficient targeted siRNA delivery with robust therapeutic efficacy for cancer treatment. The potent efficacy of siRNA/Ap-CS was also validated by the cell apoptosis assay via the flow cytometry based on the Annexin V-FITC/PI staining. As shown in Supplementary Fig. 20, compared with the viable cells incubated with PBS with 99.6% cell viability (panel 1), the apoptotic cell population of MCF-7 cells treated with siRNA/Ap-CS is 40% (panel 4), including the apoptotic cells in the late and early phase (Q2 + Q3). This value is substantially higher than the naked siRNA-treated cell group (9.1% = 5.3% + 3.8%, panel 2), even higher than the cell group treated with Lipo3000/siRNA (13.3% = 7.5% + 5.8%, panel 3), indicating that siRNA/Ap-CS is a potential RNAi therapeutic platform superior to the effective

transfection agent available commercially. In addition, MCF-7 cells were further stained with calcein-AM and PI for confocal fluorescence imaging. As shown in Supplementary Fig. 21, the cells treated with siRNA/Ap-CS display the higher red fluorescence signal than other control groups, evidently indicating the death of more cells. Supplementary Fig. 22 demonstrates that RNAi response does originate from siRNA/Ap-CS formulation because no change is observed in *Plk1* mRNA expression, cell viability and cell apoptosis if scrambled sequence was used instead of siPlk1. Similarly, if the unreleasable siRNA was instead employed, the RNAi effect is compromised. Besides MCF-7 cell line, one type of breast cancer cells, Supplementary Fig. 23 describes the RNAi potency of siRNA/Ap-CS in A549 cells, demonstrating the developed siRNA delivery vehicles are suitable for the therapeutic intervention of other types of tumor cells, e.g., adenocarcinomic cell line. Moreover, we also used the developed nano-vehicle to deliver the siRNA capable specifically silencing caspase-3 that causes rat insulinoma (INS-1E) cell death. The corresponding siRNA-incorporated formulation is called siCA3/Ap-CS. As shown in Supplementary Fig. 24, after treated with siCA3/Ap-CS, over 59% caspase-3 miRNA expression in the cells was suppressed, and the cell viability increases from 61% to 100%. In contrast, scrambled siCA3/Ap-CS cannot induce the significant change in miRNA expression and cell viability. The experimental results indicate the generality of this delivery system.

**In vivo therapeutic efficacy for NSCLC treatment**. Encouraged by effective gene silencing and therapeutic efficacy in vitro, we

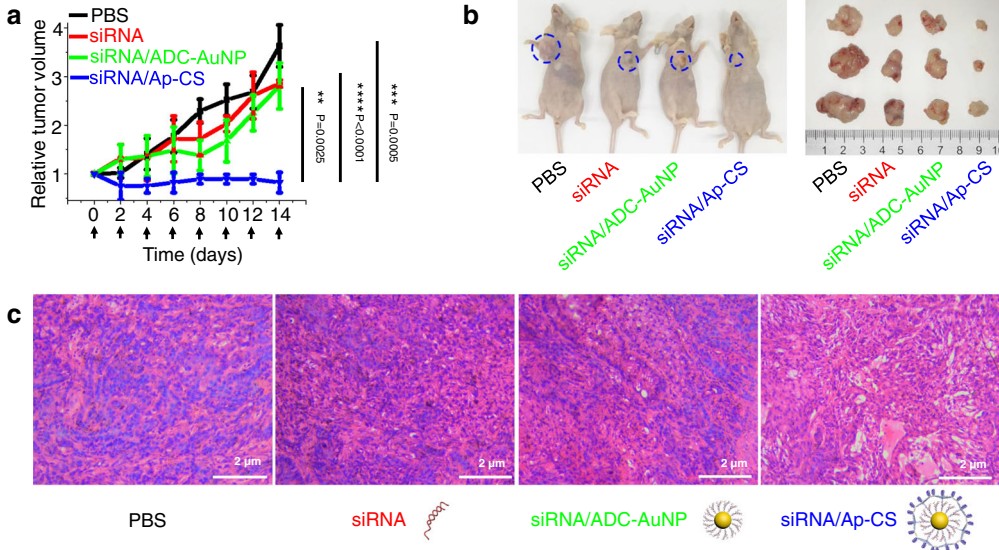

**Fig. 7 In vivo therapeutic efficacy of siRNA/Ap-CS against malignant tumor in A549 NSCLC xenograft murine model. a** Tumor growth curves during treating with siPlk1 delivered by different formulations. The arrows represent the intravenous injections via tail vein. The error bar represents the standard deviation (SD). The measured data are expressed as the means ± SD ($n = 3$). **b** Images of tumor-bearing mouse and corresponding harvested tumors 14 days after treatment with different siPlk1-encapsulated formulations. **c** Hematoxylin and eosin (H&E) staining-based histological images of tumor sections obtained from the same model mice as **b**. The histological examination was performed three times independently with similar results. **$P < 0.01$, ***$P < 0.001$, ****$P < 0.0001$, two-tailed unpaired $t$ test.

further evaluated the in vivo performance of siRNA/Ap-CS in cancer therapy. Immunodeficient mice bearing A549 (human NSCLC) tumor xenograft were randomly divided into 4 groups and siPlk1-contained formulations, including siPlk1/Ap-CS, were administered into the mice through the tail vein at the same time every 2 days up to 14 days. The tumor size was measured with digital calipers before each siRNA treatment, and the relative tumor volume is presented in Fig. 7a, showing that siPlk1/Ap-CS almost completely inhibited tumor growth compared with the three control groups. Figure 7b demonstrates that substantial tumor growth inhibition by siPlk1/Ap-CS can be directly observed by the naked eye. Furthermore, the therapeutic efficacy of siPlk1/Ap-CS in mouse models was also evidenced by the comparative histology analyses based on hematoxylin and eosin (H&E) staining of tumor tissue section. As presented in Fig. 7c, the tumor section treated with siRNA or siRNA/ADC-AuNP shows a well-defined nucleus and cytoplasm similar to the PBS-treated group. In contrast, the nuclear atrophy and extracellular space increment are observed in the siRNA/Ap-CS-treated group, indicating the more efficient cell apoptosis in tumors[57]. In addition, no systemic toxicity occurs during the treatment because no loss of body weight is detected for each group of model mice as shown in Supplementary Fig. 25. Besides, unlike the three control group very often with anxiety-like behavior and scratching action, the group administered with siRNA/Ap-CS has no obvious abnormal behaviors, including eating, drinking, activity, defecation and urination, throughout the period of examination. Moreover, in vivo tumor accumulation and therapeutic efficacy of siRNA/Ap-CS were evaluated by comparing with scrambled siRNA/Ap-CS, siRNA/CS, and Liposome/siRNA. As shown in Supplementary Fig. 26, siRNA/Ap-CS displays the superior in vivo tumor-targeting ability and therapeutic efficacy. These experimental results demonstrate that siPlk1s can be delivered to target cells in the tumor tissues by the as-developed core/shell nano-vehicle with active tumor-targeting capability and knockdown PLK1 protein, inhibit cell proliferation, induce cell apoptosis, and eventually suppress tumor progression.

## Discussion
In summary, an efficient AuNP/oligonucleotides core/shell-type siRNA delivery nanosystem, siRNA/Ap-CS, has been developed via tiling sticky Ap-YTDB on siRNA-loaded AuNP surface to form a protective outer layer with active tumor-targeting capability, circumventing the two functional limitations: the susceptibility to enzymatic degradation in vivo and nonspecific cell permeability, encountered by emerging SNA-based delivery platforms. In essence, sticky Ap-YTDB is a label-free multi-functional triangular DNA brick harboring Y-shaped dsDNA backbone in the center, sticky ends and vertex-confined aptamers, providing a structural rigidity capable of enhancing nuclease resistance and the ability to specifically recognize cell surface receptors. On the nanoparticle surface, the triangular DNA brick units are able to self-assemble into a well-organized and cross-linked DNA coating, endowing the siRNA-encapsulated formulation with high serum stability and cancer-specific cellular internalization without any chemical modification. Thus, besides the high siRNA payload capacity, siRNA/Ap-CS possesses the longer blood circulation lifetime, significantly delaying the systemic clearance and thus ensuring the sufficient time for the encapsulated siRNAs to reach tumor tissues and accomplish the escape from the endosome into cytoplasm. Moreover, the loaded siRNAs are easily designed to be released by cancer biomarkers, for example, intramolecular miRNAs. As a result, the loaded siRNAs are directly delivered and fully released into the intracellular environment of tumor, leading to the effective silencing of target gene and the remarkable suppression of cancer-related protein expression. This induces cell apoptosis and substantially inhibits tumor growth. Additionally, when DNA building blocks or siRNAs are fluorescently modified, the in vivo biological processes (e.g., intracellular and extracellular behaviors, biodistribution in vivo and pharmacokinetic properties) can be screened in real-time manner and therapeutic response assessment is able to be conveniently monitored, yielding valuable insight into the integration of diagnostic and therapeutic functions into a theranostic formulation. All the experimental results achieved at the cellular level, at the tissue level and at the level of

the whole organism, including targeted delivery and controlled release of siRNAs, gene silencing, knockdown of target protein expression and tumor growth suppression, show that the developed core/shell formulation holds remarkable potential as an intelligent siRNA delivery system responsive to intracellular molecular triggers for targeted cancer theranostics.

## Methods

**Materials**. DNA oligonucleotides listed in Supplementary Table 1 were synthesized by Sangon Biotech Co., Ltd. (Shanghai, China). Label-free and modified DNAs were purified by native polyacrylamide gel electrophoresis (PAGE) and HPLC, respectively. Human breast cancer cell lines (MCF-7), human cervical cancer cells (HeLa) and human lung cancer cells (A549) were obtained from Institute of Biochemistry and Cell Biology, Chinese Academy of Science (Shanghai, China) and cultured in Dulbecco's modified Eagle's medium (DMEM, high glucose) containing 10% Fetal bovine serum (FBS) and 1% penicillin-streptomycin. $HAuCl_4$ and trisodium citrate were purchased from Sinopharm Chemical Reagent Co. (Shanghai, China). Hoechst 33342 and $DEPC-H_2O$ were bought from Beijing DingGuo Biotechnology Co. (Beijing, China). Lipofectamine 3000 reagent was purchased from Thermo Fisher (Waltham, MA, U.S.A.). Annexin V-FITC and propidium iodide (PI) were purchased from Genview (Gen-view, scientific Inc., USA), while calcein-AM and cell counting kit-8 (CCK-8) were provided by Bestbio (Shanghai, China). Other chemical reagents (analytical grade) were employed as provided. All solutions and deionized water were treated with 0.1% diethylpyrocarbonate (DEPC) followed by autoclaving at 121 °C for 60 min before further use.

### Assembly of siRNA/Ap-CS

*Synthesis of AuNPs.* The monodisperse suspension of AuNPs was prepared according to the previous procedure with some modifications[58]. Briefly, 50 mL of $10^{-2}$% (w/w) $HAuCl_4$ was first heated to boiling under stirring, followed rapidly by addition of freshly prepared trisodium citrate (1% w/w, 0.5 mL). Then, the solution color changed to red. After kept under boiling and stirring for 20 min, the resulting solution was cooled down to room temperature. The prepared AuNPs were stored at 4 °C before use.

*Functionalization of AuNPs.* Thiolated ADC-modified AuNPs were synthesized according to the literature method with slight modifications[18]. Briefly, ADC (31 µL, 100 µM) was first reduced by 4 µL of 10 mM tris (2-carboxyethyl) phosphine hydrochloride at room temperature for 1 h. After mixing with 1 mL of AuNPs, the solution was kept for at least 16 h under shaking. Subsequently, phosphate buffer (100 mM $Na_2HPO_4$ and 100 mM $NaH_2PO_4$, pH = 7.4) and NaCl (2 M) were gradually added to achieve the final concentration of 10 mM and 0.1 M, respectively, followed by 8-h incubation. Afterwards, NaCl (2 M) was further added to obtain the final concentration of 0.2 M. After another 8-h incubation, NaCl (2 M) was added again to reach the final concentration of 0.3 M. Finally, after centrifuging at 12900 g for 25 min, AuNPs were washed and resuspended in 1 mL of Tris-HCl buffer (20 mM, 100 mM NaCl, 5 mM $Mg^{2+}$, pH = 7.4), achieving the expected ADC-AuNP.

*Quantification of anchoring DNAs covalently attached to AuNPs.* Dithiothreitol (DTT, 1 M) was added to ADC-AuNP (1 mL) to reach the final concentration of 0.5 mM and reacted for at least 24 h. The displaced ADCs from AuNPs were collected by centrifuging at 12900 g for 25 min and quantified by Q5000 UV/Vis spectrophotometry (Quawell, USA) at 260 nm. The concentration of AuNPs (0.69 nM) was estimated by calculating the ratio of the absorbance value of AuNPs at the surface plasma resonance peak ($A_{spr}$) and absorbance value at 450 nm ($A_{450}$)[59]. Based on these measured data, we could estimate the ratio of covalently attached ADC-to-AuNP: the amount of DNA is about 720 ADCs per AuNP.

*SiRNA loading.* Excess siRNA duplex (siRNA silencing luciferase, named siLuc; siRNA silencing *Plk1* gene, named siPlk1) were first prepared by mixing equal amount (10 µL, 10 µM) of sense strand and antisense strand in 30 µL of PBS (NaCl 137 mmol/L, KCl 2.7 mmol/L, $Na_2HPO_4$ 10 mmol/L, $KH_2PO_4$ 2 mmol/L, pH 7.4). Then, the resulting siRNA solution was added to ADC-AuNP (200 µL) and allowed to react for at least 2 h. The residual siRNAs were removed by centrifuging at 12900 g for 25 min. The sediment was resuspended in 200 µL of Tris-HCl buffer (20 mM, 100 mM NaCl, 5 mM $Mg^{2+}$, pH = 7.4), finally forming siRNA/ADC-AuNP.

*Construction of siRNA/Ap-CS.* YTDB-a was first assembled according to the following method: Equal amounts (12 µL, 10 µM) of strand 1 (S1), strand 2 (S2), and strand 3 (S3) were added into 92 µL of DEPC-treated PBS and then heated at 90 °C for 5 min. After gradually cooling to room temperature, 36 µL of linker strand a (Ls-a, 10 µM) and 36 µL of AS1411 (10 µM) were added and allowed to react for 1 h at room temperature. Afterwards, the resulting solution was mixed with 200 µL of siRNA/ADC-AuNP, followed by 5-h incubation. Finally, the siRNA/Ap-CS nanoparticles were washed twice with PBS containing 0.01% Tween (vol/vol) (t-PBS) by centrifuging at 12,900 g for 25 min and resuspended in t-PBS to a final

volume of 200 µL. The siRNA concentration in siRNA/Ap-CS solution was calculated as about 230 nM according to two facts: 420 siRNAs per AuNP and the AuNP concentration of 0.54 nM. By the way, YTDB or YTDB-a mentioned in the text is the expected sticky-YTDB unless otherwise indicated.

**Atomic force microscopy (AFM)**. YTDB-a was first assembled by mixing equal amounts (1 µL, 10 µM) of S1, S2, and S3 into 19 µL of PBS and heating at 90 °C for 5 min. After gradually cooling to room temperature, equal amount (3 µL, 10 µM) of Ls-a was added and then allowed to react for at least 1 h at room temperature. The assembly of YTDB-b and YTDB-c was conducted according to the same procedure as YTDB-a, but different groups of DNA sequences were used. Specifically, S1, S2, S3, and Ls-b were used for the preparation of YTDB-b; S2, S3, S4, and Ls-c were used for YTDB-c. The siRNA/Ap-CS was assembled as described in the section of "Assembly of siRNA/Ap-CS". After the deposition of the samples (2 µL each) onto the freshly cleaved micas for 20 min, the resulting mica surfaces were carefully washed by $H_2O$ and dried with pure nitrogen. The AFM scanning was conducted on ScanAsyst Mode on a MultiMode 8 atomic force microscope (Bruker, Germany) and the data were analyzed by Nano Scope Analysis 1.70.

**Dynamic light scattering (DLS)**. A 500-µL aliquot of nucleic acid-modified AuNPs was prepared according to the section of "Assembly of siRNA/Ap-CS". After diluting with 500 µL of PBS, the DLS measurement was conducted on Zetasizer Nano ZS90 (Malvern) with a He-Ne laser. The scattering angle was set as 90°.

**Agarose gel analysis**. Samples for gel electrophoresis assay were prepared by mixing 10 µL of naked AuNP or nucleic acid-modified AuNP solutions with of the mixture (5 µL) of glycerin and EDTA (50 mM) at the volume ratio of 4:1[60]. Agarose gel (1%) was freshly prepared. After loading into gel wells, the electrophoresis experiment was performed at a constant voltage of 80 V on an electrophoresis instrument (BIO-RAD, USA), and 0.5 × TBE (4.5 mM Tris, 4.5 mM boric acid, 0.1 mM EDTA, pH 7.9) was used as working buffer. The gel image was analyzed by Image Lab 5.0.

**Retention (%) assay of siRNA on AuNPs after treatment with 10% FBS**. In this section, anchoring DNA (ADC) and siRNA duplex (siLuc) of LS-ADC/LA-FAM (LA modified with FAM) were used, and the siRNA/ADC-AuNP nanoparticles without and with YTDB (YTDB-a, YTDB-b or YTDB-c) outermost layer were prepared according to the section of "Assembly of siRNA/Ap-CS". A 120-µL aliquot of as-prepared siRNA-incorporated nanostructure sample was mixed with equal amount of degradation DMEM (d-DMEM, 120 µL) containing 20% FBS and 2 × TAMg buffer (90 mM Tris and 25.2 mM $Mg(OAc)_2·6H_2O$, pH 8) (Step 1 in Fig. 3a). After reacting at 37 °C for a given period of time under stirring at 400 rpm, the resulting solution was centrifuged to remove the degraded siRNAs (step 2). The intact siRNAs on AuNPs were collected after 20-min treatment with urea (the final concentration of 8 M) at 45 °C under stirring at 400 rpm (step 3) and centrifugation at 12,900 g for 25 min (step 4). The fluorescence spectra of supernatant solution were measured at room temperature on F-7000 fluorescence spectrometer (Hitachi, Japan). The residual quantity of intact siRNAs was estimated from the fluorescence intensity of supernatant solution, and the fluorescence intensity of the corresponding sample without FBS treatment was defined as 100%. The arbitrary unit was used as the unit of fluorescence.

**Biodistribution kinetics of siRNA/Ap-CS and its tumor localization**. To establish NSCLC xenograft tumor model[61], about 20 g of female BALB/c nude mice were selected and subcutaneously inoculated with A549 cells ($1 \times 10^7$ cells per mouse). After the tumor grew to about 6 mm in diameter, 400 µL of siPlk1-incorportaed formulation was concentrate to 100 µL (the siRNA concentration is 0.91 µM) and intravenously injected via tail vein, and the fluorescence images were taken at the time points of 30, 60, and 90 min. PBS was used as blank, while siRNA and siRNA/ADC-AuNP served as the controls. All the whole-body imaging experiments were carried out under identical conditions. Then, the mice were killed and the organs were harvested, followed by the immediate fluorescence imaging (IVIS Living image 4.4).

### Cell internalization
*Cell Culture.* MCF-7 cells were cultured in DMEM medium containing 10% fetal bovine serum and 1% penicillin–streptomycin. The cells were kept at 37 °C in a humidified atmosphere of 5% $CO_2$.

### Cells imaging
*Experimental details.* 2000 µL of siRNA/Ap-CS was prepared as described in the section of "Assembly of siRNA/Ap-CS" and then concentrated to 200 µL. However, PA-Cy5 (*Plk1* antisense modified with Cy5) and *Plk1* sense were employed to prepare siRNA duplex. ADC, one fragment of which is complementary to miRNA, was used for the preparation of siRNA-incorporated formulation to design the siRNA "Release" group, while anchoring DNA (AD strand), incapable of hybridizing with intracellular miRNA, was adopted when assembling siRNA-

incorporated formulation to achieve the siRNA "Not release" group. To obtain the confocal cell images, the cells were firstly plated on a cover glass (22-mm) in a plastic-bottom plate (12-well) and cultured for 24 h. Afterwards, the cells were incubated with siRNA/Ap-CS, which was pre-diluted with DMEM to a final volume of 400 μL, for 4 h at 37 °C. After washing twice with PBS, the cells were treated with Hoechst to execute nucleus staining. The cell imaging was carried out on Leica SP8 laser scanning confocal microscope (Leica Microsystems CMS GmbH 1.1.0, Leica, Germany). The excitation wavelengths of 637 nm and 405 nm were used for Cy5 and for Hoechst 33342, respectively.

*Flow cytometry*. SiPlk1-incorporated formulations (1000 μL, 227 nM) were pre-pared as described in the section of "Assembly of siRNA/Ap-CS", but siPlk1 consisting of *Plk1* sense and Cy5-labeled *Plk1* antisense (PA-Cy5) was instead used. Before use, the siPlk-incorporated formulation was diluted with DMEM to a final volume of 2000 μL (the final concentration of siRNA is about 114 nM). The cells were plated on a plastic-bottom plate (six-well) and cultured for 24 h. After incubating in siPlk-incorporated formulation solution at 37 °C for 4 h, the cells were treated with trypsinization, followed by flow cytometric analysis on Becton Dickinson multiparametric fluorescence-activated cell sorting Aria III cell sorter (Flow Jo 7.6.1).

### Gene silencing performance of siRNA/Ap-CS formulation in vitro
*QPCR assay*. SiRNA/Ap-CS was prepared as described in the section of "Assembly of siRNA/Ap-CS". Various siPlk1-incorporated formulations (200 μL, 227 nM) were diluted with DMEM to the final volume of 400 μL (the final concentration of siRNA is about 114 nM) and then incubated with MCF-7 cells for 48 h. For qPCR analysis, total RNAs of as-treated MCF-7 cells were extracted using Trizol Reagent Kit (Invitrogen). Reverse transcription of mRNA and qPCR experiments were carried out using PrimeScript™ RT reagent Kit with gDNA Eraser (Takara, Dalian, China) and TB Green™ Premix Ex Taq™ (Takara, Dalian, China), respectively. QPCR was performed by using Bio Rad CFX Manager 3.0. The expression level of mRNA was quantified by the $2^{-\Delta\Delta Ct}$ (threshold cycle) method.

*Western blotting*. MCF-7 cells were separately treated with various siPlk1-incorporated formulations for 48 h. After washing by PBS, the cells were lysed by the mixture of RIPA buffer and PMSF (Phenylmethanesulfonyl fluoride) at the ratio of 100:1. RIPA buffer and PMSF were all bought from Beijing Ding Guo Biotechnology Co., Beijing, China. The total proteins in each sample were quan-tified by BCA Protein Assay kit (Takara, Dalian, China) and separated by a 10% SDS-PAGE. Afterward, the samples were transferred to nitrocellulose membranes (NC membranes) through electrophoretic blotting. The membranes were sealed with nonfat milk powder and immunoblotted with primary antibody against *Plk1* (Rabbit mAb #4513, Cell Signaling Technology) at the dilution of 1:1000. GAPDH (Rabbit mAb #5174) was used as the endogenous control and Anti-rabbit IgG (HRP-linked Antibody) #7074 was used as the secondary antibody.

*Cell viability*. After incubation with the siPlk1-incorporated formulation for 48 h, the viability of MCF-7 cells were analyzed by a CCK-8 assay according to the manufacturer's instruction.

### Gene silencing performance of siRNA/Ap-CS formulation in vivo.
To establish NSCLC xenograft tumor model[61], about 20 g of female BALB/c nude mice (kept at an ambient temperature of 25 °C and a relative humidity of 40–60% under a 12-h light/dark cycle) were selected and subcutaneously inoculated with A549 cells (1 × 10⁷ cells per mouse). After the tumor grew to about 6 mm in diameter, 400 μL of siPlk1-incorportaed formulation was concentrate to 100 μL (the siRNA con-centration is 0.91 μM) and intravenously injected via tail vein every 2 days (the siRNA dose used for in vivo studies is about 0.1 mg/kg). After two-week treatment, the tumor was harvested and analyzed by histological section.

The in vivo therapeutic efficacy of siRNA/Ap-CS formulation was investigated using A549 (human NSCLC) tumor xenograft tumor model. Tumor-bearing mice were randomly divided into four treatment groups (4 per group). When the tumor size reached 6 mm in diameter, different formulations including (i) PBS, (ii) siPlk1 alone, (iii) siPlk1/ADC-CS and (iv) siPlk1/Ap-CS were separately administrated intravenously at a dose of 100 μL of siRNA (the siRNA concentration is 0.91 μM) per animal every other day. Tumor size was measured before the next injection every other day by caliper. The tumor growth was estimated from the relative tumor volume of $V/V_0$, where $V$ and $V_0$ were the tumor volume after and before treatment, respectively. The tumor volume was calculated by the formula = tumor length × (tumor width)²/2.

For histological analysis, following 14-d treatment, the A549 tumor was harvested and fixed in 4% paraformaldehyde. After embedded in paraffin, the 2-to-3-μm-thick histological sections were cut and stained with hematoxylin (blue fluorescence) and eosin (red fluorescence) (H&E), followed by the examination by histopathology. The animal experiment was approved by the Institutional Animal Care and Use Committee (IACUC) of Fuzhou University (approval number: SYXK-2019-0007).

**Statistical analysis**. GraphPad Prism 8 was used for the statistical analysis in this paper. Statistical significance was determined using a two-tailed unpaired *t* test. The results were presented as mean ± standard deviation. $P < 0.05$ was considered to be statistically significant.

**Reporting summary**. Further information on research design is available in the Nature Research Reporting Summary linked to this article.

## Data availability
The data are available from the corresponding authors upon reasonable request. Source data are provided as a Source Data file.

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

## Acknowledgements

This work was supported by National Natural Science Foundation of China (21775024) and Natural Science Foundation of Fujian Province (2019J02005).

## Author contributions

Z.S.W. conceptually designed the experimental strategy, provided intellectual input, supervised the project, and wrote the paper. C.X. designed the detailed experiments, performed the experiments, analyzed the data and helped to write the paper. S.H. and Z.H.G. performed the preparation of the nanoparticles and the cell experiments. L.W., X.Y., and B.L. performed the animal experiments. M.X.L. performed western blotting and PCR experiments. Z.F.S. helped analyzed the data.

## Competing interests

The authors declare no competing interests.

## Additional information

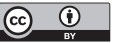

