## [Peer Review File · Nature Communications]

REVIEWER COMMENTS

Reviewer #1 (Remarks to the Author):

In this work, the authors developed a siRNA delivery system where the siRNA is hybridized to a complementary thiol-conjugated DNA strand covalently attached to gold nanoparticles and covered with aptamer-incorporated, Y-shaped backbone-rigidified triangular DNA bricks. After synthesis and characterization of the nanoconstructs, the authors demonstrate their ability to use these structures to target cancer cells expressing the aptamer binding domain and appropriately release siRNA through endogenous miRNA binding and displacement. Once the siRNA has been released from the nanoparticle, in vitro gene knockdown occurs, which is investigated by looking at mRNA expression and Western blots of protein levels of PLK1, a regulator of mitotic progression in mammalian cells that is overexpressed in cancer cells. In addition, the authors apply this system to an in vivo xenograft model, where they demonstrate, after 14 days, a ~three-fold reduction in tumor volume compared to the siRNA treatment alone.

There are some issues below that should be addressed before I can reconsider this article for publication. Overall, this paper needs major revisions. There are many claims made in this work that would be very novel and exciting - if only the authors had all the data needed to support them. Otherwise, the authors need to adjust their claims because the data does not conclusively show everything that they are claiming.

- 1) Page 5: The authors' use the term ADC many times, but never appropriately define it upon its initial mention on this page, but rather define it in Figure 1.
- 2) Page 11: The authors say that Figure 2C shows a gel of the electrophoretic mobility of the different constructs, but the gel image is multi-colored and has both blue and purple bands. These bands are never defined for the reader, and thus it is very difficult to understand the conclusions from this figure.
- 3) Page 12: The authors say that the red shift in the UV-vis spectrum in Figure 2E is indicative of the successful assembly of the nanoparticles, but this is unclear why this shift demonstrates that and no explanation is provided as to how they have made this claim.
- 4) Page 12: In Figure 3B, the legend (and accompanying text in the paper) says, "Samples a-e" but the authors never tell the reader what each sample corresponds to. It would be significantly more helpful if the authors would list the sample names in the key.
- 5) Page 13: The authors say that the percentage of residual siRNA in their YTDB-a-based particle is larger than the double protection offered and cite chemical modification as one of the two methods of protection. However, it was my understanding that they don't use a chemical modification method in this paper, and they never provide the appropriate controlled comparison in Figure 3 to demonstrate how each of the components of protection (SNA and chemical modifications) add to the stability of their nanoparticle.
- 6) Page 15: The authors have a section in the results titled, "Targeted Delivery, Controlled Release and Gene Silencing Activity." However, in this section, all of the data is in the SI. This data is critically

important to the authors' claims that their NP is able to target the delivery of siRNA in vivo and not having any of this data in the main text is detrimental to the novelty and impact of this work.

7) Page 15: The authors say that, "The stimulus-responsive release of siRNA can be realized because the ADC is capable of preferentially hybridizing with endogenous miRNA-21." This claim is not true, and the binding is not "preferential" because the authors do not provide any data with other miRNA strands to indicate that it is specific to miRNA-21. miRNAs are notoriously non-specific related to binding to their targets, and many have hairpins that indicate that it is only a handful of the bases within the entire strand that are perfectly complementary to their target. It is for this reason that miRNA is able to bind to multiple targets and can affect the expression of multiple different genes. The authors also should provide data on the binding of other types of oligonucleotides (such as other siRNA or DNA) to indicate that the specificity is truly for miRNA-21, if they wish to make this claim.

8) Page 15: The authors should better clarify how the aptamer is incorporated or related to the Y-DNA brick, as it appears that after attaching aptamer AS1411, they would block the Y-shaped backbone-rigidified triangular DNA bricks that enable their system to work.

9) Page 16: The authors indicate that 73% of the genes were silenced but should better indicate in the main text how they calculated this number so that the reader does not have to go to the SI to see this piece of data. In addition, in comparing the system without aptamer to the one with AS1411 only, a ~10% reduction (according to figure S12) is seen, and no benefit is seen compared to the transfection reagent. The authors need to provide more context for this data in the main text, as it does not indicate a significant improvement.

10) Page 16: The authors claim that "a strong red fluorescence signal is detected in fluorescence image (Cy5) and a considerable number of AuNPs, [...] demonstrating the successful cellular uptake of siPlk1/Ap-CS and intracellular miRNA-triggered the efficient release of formulated siPlk1." This claim is not valid. The other thing that the Cy5 signal demonstrates is that there is particle uptake by the cells of their nanoparticle. It doesn't indicate that there is efficient release. Furthermore, to support their point, the authors should provide verification to indicate that there is quenching of the Cy5 until the miRNA binds and releases it, at which point they can make the claim written in the text.

11) Page 16: Figure 4B shows confocal images and an accompanying flow plot. How representative are these confocal images of the entire population? I would much rather see flow data from multiple experimental replicates with appropriate error bars so that we can determine the population variability of the system. In addition, was the flow gated on live cells? How could it have been if their system was targeting a protein that would cause cell death upon delivery? The authors should use a scramble siRNA to produce this data and ensure appropriate flow cell gating.

12) Page 16: "Meanwhile, almost as many AuNPs are seen." This appears to have been done relatively by eye. Please quantify this using a more appropriate technique.

13) Page 17: The authors say that SNAs suffer from a limitation of the susceptibility to nuclease degradation. This claim is not true as the SNA is significantly less susceptible than its linear counterparts - *Nano Lett.* 9, 308-311 (2009). This paper should also be cited: <https://www.nature.com/articles/382607a0>.

14) Page 17: The authors state that their system has overcome the bottleneck of in vivo delivery to the tumor site. However, this was not appropriately demonstrated or described in Figure S7 and S8. Upon organ dissection, the authors get significantly higher liver accumulation than their controls,

which would indicate that relative percentage accumulation in tumors is lower. This is not addressed in the text, and the indicated “tumor circles” in Figure S8 do not provide a quantitative relative measure of tumor accumulation for the timescales provided.

15) Page 18: The authors need to clarify if they transfected their nanoconstructs into cells, as they state, “including fluorescence image and bright field image by laser scanning confocal microscope, siPlk1/Ap-CS-transfected cells were analyzed by flow cytometry.” I thought these systems did not require transfection vehicles.

16) Page 19: The authors need to provide more explanation and clarification as to their conclusion that “for the un-siPlk1/Ap-CS where siRNA cannot be displaced by intracellular miRNA and its fluorescence is partly quenched by AuNP, the treated cells only offer a moderate optical signal (purple line).” It is not intuitive as to why displacement is necessary to measure uptake, and this would answer an earlier question, where an experiment demonstrating quenching upon attachment and release should be provided to the reader.

17) Page 19: There are a few questions regarding the claim that: “Moreover, the internalized siRNA/Ap-CS can successfully escape from endosome/lysosome into the cytosol (SI Appendix, SI Fig. S13), naturally contributing to the efficient gene silencing.” How many cells were imaged for those quantifications? LysoTracker is nonspecific for all acidic compartments so it is hard to say at what point in the endo/lysosomal process these particles left and became non-colocalized. How do their particles compare to other controls, as none were shown in the accompanying data? How can the authors prove that it was not just the fluorophore becoming cleaved from the siRNA and that was escaping the compartments? Where is the time-course of gene knockdown that supports this data?

18) Page 19: Why did the authors select 48h as the timepoint? What about assessing qPCR expression at earlier time points? If they observe “escape” at 10h, why not try 24h or even 15h? Something closer to 10h?

19) Many times throughout the paper, the authors say things are “obvious” or “clearly” or “impressive.” These terms should be removed, and the grammar should be corrected to provide more substantiated conclusions based on the data provided.

20) Page 24: Figure 6A/B: Is there significance between the groups? Please provide those stats.

21) SI, Figure S9A: Shouldn't there be two bands in the gel - one for siRNA that was just released and another for the miRNA/ADC complex?

22) SI, Figure S11: Where are the appropriate controls that show all images of other conditions?

23) SI, Figure S17: For this figure, but also for the flow plots that are shown in general, how many replicates were run? Is this just one experimental batch?

Reviewer #2 (Remarks to the Author):

The manuscript entitled “Programmably tiling rigidified-DNA brick on AuNP as multi-functional 3D shell for cancer-targeted delivery of siRNAs” describes the construction of a spherical nucleic acid delivery system with increased serum stability and responsiveness to endogenous stimuli, and demonstrates its application for siRNA delivery in mouse xenograft model. This work addresses important challenges for the systemic application of spherical nucleic acid technology and is a valuable addition to the toolbox of SNA formulation. The authors took a novel approach to create Y-shaped backbone-rigidified triangular DNA bricks that can protect the siRNA from degradation and enable incorporation of the aptamer to enhance targeting of tumor cells. The authors also investigated the use of endogenous miRNA as a trigger to enhance the release of siRNA by a strand displacement reaction. The major criticism is related to the design and conclusions from in vivo work and lack of statistical analysis which should be addressed.

Major comments:

1. The authors should include the addition group containing scrambled siRNA to distinguish between the therapeutic efficacy of siRNA/Ap-CS and the effect of tumor accumulation of the targeted particles. siRNA/ADC-AuNP show drastically different accumulation in the tumor than targeted NPs and can't be directly compared. Demonstrating in vivo Plk-1 mRNA or protein knockdown would also help strengthen the evidence of therapeutic efficacy.
2. On page 5 authors claim that their system displays long circulation lifetime. To support this claim authors should provide the pharmacokinetic data such as plasma clearance, the steady-state volume of distribution, and half-life. Based on the data presented in Figure S8 it is not clear whether including the outer coating can influence the circulation times in vivo, which could also contribute to tumor delivery. This is important in light of the data from Figure 5, showing that enhanced stability provided by the coating does not significantly contribute to the silencing efficacy in the xenograft model.
3. The targeted siRNA/Ap-CS particles show 6-fold higher accumulation in the liver as compared with siRNA/ADC-AuNP particles (Figure S) but the authors do not provide any explanation for this observation in the manuscript and the extended discussion S3.
4. Given the strong accumulation of the siRNA/Ap-CS particles in the liver, the manuscript would greatly benefit from including the liver toxicity data (e.g. ALS, AST, AP) to support the claims on the systemic toxicity, which authors based solely on observing the changes in animal weight and behavior after treatment.
5. It is not clear how the siRNA encapsulation efficacy and the final dose were calculated for the developed system used for in vivo studies. To also allow a better comparison with other delivery modalities, the siRNA dose used for in vivo studies should be expressed in mg/kg.
6. Authors should perform appropriate statistical analysis and indicate the statistically significant difference for they're in vivo and in vitro data (Fig 5, 6, S12, S16, S17)

Minor comments:

1. On page 15: Authors cite the work of Jeong et al. *Bioconjugate Chem.* 20, 5-14 (2009) and state that “chemical modification of siRNAs compromises the silencing activity” in the context of using chemical modifications of siRNA with AuNP. This statement is misleading and should be withdrawn or corrected since neither the authors nor the cited work has shown the validity of this with AuNP and this effect will likely depend on the type of modifications used. In fact, the majority of clinically evaluated siRNAs and approved drugs like Parisian and Givosiran use modified siRNAs.

Responses to the reviewer's comments

All the authors would like to express our gratitude to the referees for their time and constructive comments. Our response to each comment is provided below following each comment raised.

Comments:

Reviewer #1 (Remarks to the Author):

In this work, the authors developed a siRNA delivery system where the siRNA is hybridized to a complementary thiol-conjugated DNA strand covalently attached to gold nanoparticles and covered with aptamer-incorporated, Y-shaped backbone-rigidified triangular DNA bricks. After synthesis and characterization of the nanoconstructs, the authors demonstrate their ability to use these structures to target cancer cells expressing the aptamer binding domain and appropriately release siRNA through endogenous miRNA binding and displacement. Once the siRNA has been released from the nanoparticle, in vitro gene knockdown occurs, which is investigated by looking at mRNA expression and Western blots of protein levels of PLK1, a regulator of mitotic progression in mammalian cells that is overexpressed in cancer cells. In addition, the authors apply this system to an in vivo xenograft model, where they demonstrate, after 14 days, a ~ three-fold reduction in tumor volume compared to the siRNA treatment alone.

There are some issues below that should be addressed before I can reconsider this article for publication. Overall, this paper needs major revisions. There are many claims made in this work that would be very novel and exciting - if only the authors had all the data needed to support them. Otherwise, the authors need to adjust their claims because the data does not conclusively show everything that they are claiming.

Our response: Thanks for your time and valuable comments that help us to improve our manuscript. To address the issues raised, the corresponding experiments have been performed, and the new measured data have been offered in the revised manuscript, for example, Figures 4B-D, S5, S9, S10, S12, S14C, S14D, S16, S17, S18B, S19, S24 and S26, including some important information (e.g., comparative study of the FBS stability of siRNA/Ap-CS with the counterpart nanoformulation under double protection by SNA and 2'-OMe, and the pharmacokinetic profiles of siRNA-incorporated formulations in mice). We wish that the thorough revision make it more readable.

1) Page 5: The authors' use the term ADC many times, but never appropriately define it upon its initial mention on this page, but rather define it in Figure 1.

Our response: All the probes, including their full names, abbreviations and base sequences, are presented in Table S1. According to the suggestion, to make it easy to understand, we have defined the term ADC in page 5 by adding "anchoring DNA complementary to miRNA".

2) Page 11: The authors say that Figure 2C shows a gel of the electrophoretic mobility of the different constructs, but the gel image is multi-colored and has both blue and purple bands. These bands are never defined for the reader, and thus it is very difficult to understand the conclusions from this figure.

Our response: The red bands in Figure 2C denote the position of naked and DNA-functional AuNPs. In the revised manuscript, besides the red bands are marked by arrows, we state “The red bands indicate the naked or DNA functionalized AuNPs” in the figure caption.

3) Page 12: The authors say that the red shift in the UV-vis spectrum in Figure 2E is indicative of the successful assembly of the nanoparticles, but this is unclear why this shift demonstrates that and no explanation is provided as to how they have made this claim.

Our response: UV-vis spectroscopy is a useful technique to characterize the surface modification of AuNPs with nucleic acids because oligonucleotide functionalization can induce the change in dielectric constant of the surrounding environment of AuNP (Proceedings of the National Academy of Sciences, 2017, 114 (47), 12419–12424). Thus, it is reasonable to observe the red shift of the UV/Vis spectrum of core/shell nanoparticles compared to naked AuNPs as shown in Fig. 2E, which is consistent with literature reports (e.g., ACS Nano, 2014, 8, 12, 12386–1239; Analytic Chemistry, 2017, 89, 8377–8383).

In the revised manuscript, to make it easy to understand, we have changed “Additionally, the red-shift of the UV/Vis spectrum of core/shell nanoparticles is observed compared to naked AuNPs (Fig. 2E), indicating the successful assembly of siRNA/Ap-CS” to “Additionally, the red-shift of the UV/Vis spectrum of core/shell nanoparticles is observed compared to naked AuNPs (Fig. 2E) because the assembly of protective outer coating induces the change in the dielectric constant of surrounding environment of AuNP⁴⁸, which is consistent with literature reports^{18,48}. These experimental results indicate the successful assembly of siRNA/Ap-CS”.

4) Page 12: In Figure 3B, the legend (and accompanying text in the paper) says, “Samples a-e” but the authors never tell the reader what each sample corresponds to. It would be significantly more helpful if the authors would list the sample names in the key.

Our response: After revision, the definitions of Samples a-e are provided in the caption of Figure 3B.

5) Page 13: The authors say that the percentage of residual siRNA in their YTDB-a-based particle is larger than the double protection offered and cite chemical modification as one of the two methods of protection. However, it was my understanding that they don't use a chemical modification method in this paper, and they never provide the appropriate controlled comparison in Figure 3 to demonstrate how each of the components of protection (SNA and chemical modifications) add to the stability of their nanoparticle.

Our response: In the original manuscript, what we meant was that the FBS stability of siRNA/Ap-CS is higher than those under the double protection reported in a previous study (Proc. Natl. Acad. Sci. U.S.A. 2014, 111, 9739-9744) in which the residual content of siRNAs can be seen from the corresponding figures.

According to the suggestions raised, the siRNAs modified with 2'-OMe moieties were synthesized and loaded onto ADC-AuNP to achieve the SNA/chemical modification-based double protection. Their FBS stability was evaluated by comparing with the corresponding siRNA/Ap-CS, and the measured data are shown in Figure S5, where three siRNA-loaded formulations (Nanoparticle-a, Nanoparticle-b and Nanoparticle-c) are involved. Nanoparticle-a is the siRNA/Ap-CS with the developed protective outer coating, Nanoparticle-b is 2'-OMe-siRNA/ADC-AuNP with the double protective effect of SNA structure and 2'-OMe modification, and Nanoparticle-c is siRNA/ADC-AuNP only with the protective SNA structure. As shown in Figure S5, the FBS stability of siRNAs encapsulated in the three formulations decreases in the order of Nanoparticle-a > Nanoparticle-b > Nanoparticle-c. Thus, in the revised manuscript, we state “The residual amount of siRNAs encapsulated in the three formulations decreases in the order of Nanoparticle-a > Nanoparticle-b > Nanoparticle-c. For example, at the 60-min incubation, the residual amounts of siRNAs in Nanoparticle-a, Nanoparticle-b and Nanoparticle-c decrease to approximately 91%, 64% and 42%, respectively. These experimental data demonstrate that the FBS stability of siRNA/Ap-CS indeed is much higher than the counterpart formulation, 2'-OMe-siRNA/ADC-AuNP, with SNA/2'-OMe-based double protection”.

Fig. S5. Comparative study of the FBS stability of siRNA/Ap-CS with the counterpart nanoformulation under double protection by SNA and 2'-OMe. Quantitative evaluation of the retention (%) of surface-confined siRNAs is represented after exposure of different siRNA-encapsulated formulations (a-c) to FBS, where the retention efficiency of nontreated formulation corresponding to each group is defined as 100%. Nanoparticle-a is the siRNA/Ap-CS with the protective outer coating, Nanoparticle-b is 2'-OMe-siRNA/ADC-AuNP with the protective effect of SNA structure and 2'-OMe modification, and Nanoparticle-c is siRNA/ADC-AuNP only with protective SNA structure.

6) Page 15: The authors have a section in the results titled, “Targeted Delivery, Controlled Release and Gene Silencing Activity.” However, in this section, all of the data is in the SI. This data is critically important to the authors’ claims that their NP is able to target the delivery of siRNA in vivo and not having any of this data in the main text is detrimental to the novelty and impact of this work.

Our response: After revision, we have moved the important data (Fig. 4) to the main text.

7) Page 15: The authors say that, “The stimulus-responsive release of siRNA can be realized

because the ADC is capable of preferentially hybridizing with endogenous miRNA-21.” This claim is not true, and the binding is not “preferential” because the authors do not provide any data with other miRNA strands to indicate that it is specific to miRNA-21.

Our response: To demonstrate that ADC preferentially hybridizes with miRNA-21, we have evaluated the fluorescence emission intensity of our system in the presence of different miRNAs in a comparative manner. As shown in Figure S12A, in the presence of miRNA-21D, the highest fluorescence signal is observed, while other miRNAs (miRNA-429, miRNA-141 or miRNA-200b) cannot induce the detectable fluorescence change, indicating the specificity of the system towards miRNA-21.

Fig. S12. (A) The real-time monitoring of fluorescence emission intensity of siLuc/ADC-AuNP in the presence of different miRNAs (miRNA-21, miRNA-429, miRNA-141, and miRNA-200b). The siLuc duplex was prepared by hybridizing LS-ADC with FAM-labeled LUC-antisense (LA-FAM).

miRNAs are notoriously non-specific related to binding to their targets, and many have hairpins that indicate that it is only a handful of the bases within the entire strand that are perfectly complementary to their target. It is for this reason that miRNA is able to bind to multiple targets and can affect the expression of multiple different genes. The authors also should provide data on the binding of other types of oligonucleotides (such as other siRNA or DNA) to indicate that the specificity is truly for miRNA-21, if they wish to make this claim.

Our response: In Figure S12, besides miRNA-21 can release one siRNA (siLuc), we further provided other data on the release of another siRNA (siPlk1) and DNA (ES-ADC/EAH) to demonstrate the specificity of miRNA-21 according to the reviewer’s advice. As shown in Figures S12B and S12C, only miR-21D is able to induce the expected fluorescence signal, while all other miRNAs cannot trigger the significant fluorescence change.

Fig. S12. (B) and (C) are the same as (A), but siLuc was substituted with dsDNA and siPIK1, respectively. The dsDNA was prepared by hybridizing FAM-labeled ES-ADC to EAH, while siPIK1 was obtained by hybridization of Cy5-labeled PA with PS-ADC.

8) Page 15: The authors should better clarify how the aptamer is incorporated or related to the Y-DNA brick, as it appears that after attaching aptamer AS1411, they would block the Y-shaped backbone-rigidified triangular DNA bricks that enable their system to work.

Our response: As shown in Fig. S2B, we have offered the schematic illustration of the assembly of Ap-YTDB. Specifically, aptamer is incorporated onto Y-shaped backbone-rigidified triangular DNA brick (YTDB-a) by hybridizing to the Complementary overhang (green line). In this case, Terminal sucker of YTDB-a bricks can hybridize with the terminal complementary fragments of ADCs, and they cross-interact with each other by the palindromic Sticky ends (yellow segment) at the vertexes, forming the protective outer coating of siRNA/Ap-CS. Namely, the binding of aptamer to Y-DNA brick does not affect its function .

Fig. S2. (B) The schematic illustration of the binding of AS1411 aptamers to YTDB-a.

9) Page 16: The authors indicate that 73% of the genes were silenced but should better indicate in the main text how they calculated this number so that the reader does not have to go to the SI to see this piece of data.

Our response: According to the reviewer's suggestion, in the revised main text, "The 73% of target genes was silenced" has been changed to "If the expression level of luciferase in HeLa cells treated

with PBS is defined as 100%, only 27% luciferase is expressed after treating with siRNA/Ap-CS. Namely, about 73% of target genes is silenced...”

In addition, in comparing the system without aptamer to the one with AS1411 only, a ~10% reduction (according to figure S12) is seen, and no benefit is seen compared to the transfection reagent. The authors need to provide more context for this data in the main text, as it does not indicate a significant improvement.

Our response: Thanks for your valuable suggestions. Lipofectamine 3000 (Lipo3000) is a commercial efficient transfection reagent but without cancer cell targeting ability. Similarly, even if a small amount of siRNA/CS can enter cells that is evidenced by the appearance of several small black dots (AuNPs) in Figure 5C, the internalization is cancer cell-independent process (Proc. Natl Acad. Sci. USA, 2012, 109, 11975–11980). Thus, when being systemically administered into mice, compared with siRNA/Ap-CS, both Lipo3000/siRNA and siRNA/CS cannot efficiently accumulate in the tumor tissue, which has been confirmed as shown in Figure S26A and S26B in the revised manuscript. To assess the *in vivo* therapeutic efficacy of the three formulations in a comparative manner, they were separately systemically administered into A549 NSCLC xenograft murine models, followed by the quantitative evaluation of tumor volumes and expression level of corresponding mRNAs. As shown in Figure S26C, S26D and S26E, the *in vivo* gene silencing efficiency of siRNA/Ap-CS and the ability to suppress tumor growth are indeed much higher than Lipo3000/siRNA and siRNA/CS.

Therefore, in the Supporting information of revised manuscript, we have added the corresponding statement, “As shown in Fig. S26A and S26B, when being systemically administered into tumor-bearing mice, compared with Lipo3000/siRNA and siRNA/CS, siRNA/Ap-CS does accumulate in tumor tissues with much higher efficiency. Moreover, siRNA/Ap-CS shows the higher ability to suppress tumor growth (Fig. S26C and S26D) than Lipo3000/siRNA and siRNA/CS, which is consistent with the analysis of mRNA expression level (Fig. S26E). These data demonstrate that siRNA/Ap-CS possesses the higher *in vivo* gene silencing efficiency”.

Fig. S26. *In vivo* tumor accumulation and therapeutic efficacy of siRNA/Ap-CS against malignant tumor in A549 NSCLC xenograft murine model. (A) Time-dependent *in vivo* fluorescence imaging to explore the whole-body biodistribution kinetics of siRNA/Ap-CS and tumor localization. The samples, including scrambled siRNA/Ap-CS,

Liposome/siRNA, siRNA/CS and siRNA/Ap-CS, were administrated to the mice via tail vein injection. Cy5-labeled PA and PS-ADC were used for the preparation of siPlk1. (B) The fluorescence intensity of tumor in yellow cycles indicated in (A). (C) Tumor growth curves during treating tumor with siPlk1 delivered by different formulations. The arrows represent the intravenous injections via tail vein. The data are presented as the mean \pm SD of four independent experiments. (D) Images of harvested tumors 14 days after treatment with different siPlk1-encapsulated formulations. (E) Plk1 mRNA levels in harvested tumors were measured by qPCR. **P<0.01, ****P<0.0001, two-tailed unpaired t test.

10) Page 16: The authors claim that “a strong red fluorescence signal is detected in fluorescence image (Cy5) and a considerable number of AuNPs, [...] demonstrating the successful cellular uptake of siPlk1/Ap-CS and intracellular miRNA-triggered the efficient release of formulated siPlk1.” This claim is not valid. The other thing that the Cy5 signal demonstrates is that there is particle uptake by the cells of their nanoparticle. It doesn't indicate that there is efficient release.

Our response: We apologize for our incomplete description. In fact, almost equal amounts of siPlk1/Ap-CS and un-siPlk1/Ap-CS enter the cells because almost as many AuNPs are seen in Figure 5A as those in Figure 5B, which is also confirmed by inductively coupled mass spectrometry (ICP-MS) (Figure S16). Meanwhile, the fluorescence signal is detected from MCF-7 cells treated with siRNA/Ap-CS because siRNA can be specifically released by endogenous miRNAs (Figure S12C). But, un-siRNA is incapable of being displaced (Fig. S12E), and thus un-siRNA/Ap-CS treated cells only emit the weak fluorescence.

For more precise expression, we have now reworded “As shown in Fig. 4A, [...] cannot be released from core/shell nano-formulation within cells. Meanwhile, almost as many AuNPs are seen.” to “As shown in Fig. 5A, a strong red fluorescence signal is detected in fluorescence image (Cy5) and a considerable number of AuNPs, some of which aggregate, are observed in bright field image (BF-image), demonstrating the successful cellular uptake of siPlk1/Ap-CS and intracellular release of formulated siPlk1. The specific release of Cy5-labelled siPlk1 from the nano-formulation is verified by fluorescence measurement as shown in Fig. S12C. Unlike siPlk1/Ap-CS, Fig. 5B shows that, if un-siPlk1/Ap-CS was instead employed, the substantially compromised red fluorescence signal is detected because the un-siPlk1 cannot be efficiently released from core/shell nano-formulation within cells (Fig. S12E). But, almost as many AuNPs are seen in Fig. 5A as those in Fig. 5B, which is also confirmed by inductively coupled mass spectrometry (ICP-MS) (Fig. S16)”.

Furthermore, to support their point, the authors should provide verification to indicate that there is quenching of the Cy5 until the miRNA binds and releases it, at which point they can make the claim written in the text.

Our response: In the revised manuscript, we have added the new measured data to support the hybridization-based release of Cy5-labeled PA by real-time monitoring the fluorescence intensity of siPlk1/Ap-CS in the presence of miRNA-21 and other non-target miRNAs. As shown in Figure S12C, the Cy5 fluorescence is quenched unless miR-21D is added. Therefore, the following statement has now been added: “The specific release of Cy5-labelled siPlk1 from the nano-formulation is verified by fluorescence measurement as shown in Fig. S12C”.

11) Page 16: Figure 4B shows confocal images and an accompanying flow plot. How representative are these confocal images of the entire population? I would much rather see flow data from multiple

experimental replicates with appropriate error bars so that we can determine the population variability of the system.

Our response: Each group of laser confocal imaging experiments was independently performed three times. Moreover, we performed flow cytometry analysis that is considered to be capable of collect fluorescence data for a large population of cells (*J. Am. Chem. Soc.* 2007, 129, 50, 15477–15479). The responding quantitative data are described in the revised Figure 5F with error bars representing the standard deviation of triplicate experiments. We believe that these data, including confocal images and flow cytometry analyses, can represent the entire population.

In addition, was the flow gated on live cells?

Our response: Yes, the flow was gated on live cells. The cell viability was tested under identical conditions, and the results are shown in Figure S19. The cell viability at 4 h is approximately 98 %.

Fig. S19. (B) The corresponding cell viability. Although Plk1 mRNA level decreases with the incubation time after 15 h, the high therapeutic efficacy is achieved at 48-h incubation. Thus, the 48-h incubation was adopted in the subsequent experiments.

How could it have been if their system was targeting a protein that would cause cell death upon delivery? The authors should use a scramble siRNA to produce this data and ensure appropriate flow cell gating.

Our response: To act on the advice, we used our nano-formulation to deliver a siRNA that was designed to silence the caspase-3 capable of causing rat insulinoma (INS-1E) cells death. As shown in Figure S24, after treating the INS-1E cells with inflammatory cytokines (25ng/mL of IL-1 β , 5ng/mL of TNF- α and 25ng/mL of IFN- γ), the caspase-3 expression was increased, leading to the decreased cell death. However, after treated the resulting cells with siRNA/Ap-CS, the caspase-3 expression was inhibited, while cell viability was increased. In contrast, the scrambled siRNA/Ap-CS was unable to induce the significant change in both caspase-3 expression and cell viability.

Moreover, the comparative study of cellular internalization between siRNA/Ap-CS and scramble siRNA /Ap-CS was carried out by cell imaging and flow cytometric analysis. The resulting data are described in Figure 5 and Figure S17, respectively. One can notice that there is no substantial difference in the cellular internalization between siRNA/Ap-CS and scramble siRNA /Ap-CS.

Fig. S24. The suitability of siRNA/Ap-CS system for silencing caspase-3 protein that causes cell death. After being treated with inflammatory cytokines (25 ng/mL of IL-1 β , 5 ng/mL of TNF- α and 25 ng/mL of IFN- γ), the cells were further treated with siRNA/Ap-CS. The relative caspase-3 mRNA levels in rat insulinoma (INS-1E) cells were measured by qPCR (A), and the cell viability of INS-1E cells were test by CCK8-kit (B). The CS-ADC and CA were used for the preparation of siRNA duplex, and the resulting formulation was called siCA3/Ap-CS.

Fig. S17. Colocalization assay of scrambled siRNA (ssiRNA, red fluorescence) and AuNPs (black dot) within target cancer cells. MCF-7 cells were separately incubated with releasable siRNA/Ap-CS (A), unreleasable siPlk1 (un-siRNA)/Ap-CS (B) and releasable siRNA/CS without aptamer (C) for 4 h. (D) is the same as (A) but L02 cells were instead used. HM in the right half part is the high-resolution image of the area in yellow dotted box indicated in the section of Merge, while HBF is the high-resolution image boxed in bright field (BF). AuNPs in HM and HBF

are highlighted by yellow dotted circles. (E) Flow cytometry analysis of MCF-7 cells treated with various formulations for 4 h. (F) The quantitative fluorescence intensity of each samples in E. The measured data are expressed as the means \pm SD of three independent experiments.

12) Page 16: “Meanwhile, almost as many AuNPs are seen.” This appears to have been done relatively by eye. Please quantify this using a more appropriate technique.

Our response: In the revised manuscript, we have further confirmed the cellular internalization efficiency of siRNA/Ap-CS and un-siRNA/Ap-CS by using inductively coupled mass spectrometry (ICP-MS) to detect the concentration of AuNP per cells. As shown in Figure S16, there is no substantial difference in the concentration of AuNP between siRNA/Ap-CS and un-siRNA/Ap-CS.

Fig. S16. The concentration of AuNP per cell in Fig. 5A-D was measured by inductively coupled plasma-mass spectrometry (ICP-MS).

13) Page 17: The authors say that SNAs suffer from a limitation of the susceptibility to nuclease degradation. This claim is not true as the SNA is significantly less susceptible than its linear counterparts - Nano Lett. 9, 308-311 (2009). This paper should also be cited: <https://www.nature.com/articles/382607a0>.

Our response: We agree to the assessment that the SNA is significantly less susceptible than its linear counterparts. However, the stability of SNA is not high enough to meet all the requirements for different practical applications. For example, SNAs were reported to be degraded after 30-min incubation in FBS (PNAS, 2014, 111, 9739-9744). To offer the directly-related data on the degradation resistance of SNA-protected siRNAs designed in our study, the residual amount of siRNA/ADC-AuNP treated with 10% FBS was quantitatively estimated by fluorescence measurements. As shown in Figure S5, the residual siRNAs in siRNA/ADC-AuNP at 60-min incubation is only 42%, demonstrating the substantial enzymatic degradation. For this reason, Mirkin’s group stated “the development of SNA-siRNA with long serum lifetimes remains an outstanding challenge” (PNAS, 2014, 111, 9739-9744). Nevertheless, we have changed “the susceptibility to nuclease-induced degradation” to “the unsatisfactory resistance to nuclease-induced degradation in serum”. Additionally, the reference mentioned has been added in the main text.

14) Page 17: The authors state that their system has overcome the bottleneck of in vivo delivery to

the tumor site. However, this was not appropriately demonstrated or described in Figure S7 and S8. Upon organ dissection, the authors get significantly higher liver accumulation than their controls, which would indicate that relative percentage accumulation in tumors is lower. This is not addressed in the text, and the indicated “tumor circles” in Figure S8 do not provide a quantitative relative measure of tumor accumulation for the timescales provided.

Our response: Mirkin’s group stated “the development of SNA-siRNA with long serum lifetimes remains an outstanding challenge” (PNAS, 2014, 111, 9739–9744). To address this challenge, here we improve the stability of SNA-siRNA (here called siRNA/ADC-AuNP) by arranging a protective outer coating consisting of lying-flat aptamer-incorporated YTDB bricks, resulting in the siRNA/Ap-CS with enhanced degradation resistance besides tumor targeting ability. According to the literature report (Nat. Commun. 2017, 8, 15654), it is reasonable for siRNA/Ap-CS to accumulate in the liver when administrated into normal mice (Figure S8 in the revised manuscript) because the encapsulated siRNAs are protected from the nuclease degradation in systemic circulation. However, when administrated into the tumor-bearing mice, siRNA/Ap-CS can accumulate in tumor sites rather than livers, which is verified by *the in vivo* fluorescence images as shown in Figure 4A in the revised manuscript. Moreover, the quantitative assessment of tumor accumulation for the timescales is offered in Figure 4B, and the fluorescence images of harvested organs are provided in Figure 4C, accompanied by the quantitative assessment of siRNA/Ap-CS accumulation in liver and tumor site (Figure 4D). Apparently, the siRNA/Ap-CS can preferentially accumulate in tumor sites.

For the original Figures S7 (i.e., revised Figure S8), in the Supporting information of revised manuscript, we state: “Fig. S8A describes the biodistribution of several siRNA formulations at 1 h following post-injection into normal mice, and each group of mice show the fluorescence signal that varies in the intensity in different regions. For the siRNA/Ap-CS group, the fluorescence signal mainly focuses on the midsection. To offer accurate evaluation, the mice were killed and their organs were harvested for fluorescence imaging. As shown in Fig. S8B, all the siRNA formulations preferentially distribute in the liver rather than in other organs, but there is a significant difference in the fluorescence intensity from the liver between the three siRNA formulations. Fig. S8C shows the quantitative measurements of siRNA formulations distributed in the livers, confirming the substantial accumulation of siRNA/Ap-CS in the liver compared with naked siRNA and siRNA/ADC-AuNP. These experimental results are consistent with the previous observations¹⁶, demonstrating that siRNAs loaded in the core/shell nanostructure are protected from the nuclease degradation in systemic circulation and exhibit the enhanced stability”. Figure S9 further confirms the enzymatic stability of siRNA/Ap-CS: “The *in vivo* pharmacokinetics of siRNA/Ap-CS was also analyzed. As shown in Fig. S9, the comparative results demonstrate that siRNA/Ap-CS possesses a long blood circulation time. Specifically, its plasma half-life ($t_{1/2}$) is 2.6 and 7.1 times longer than siRNA/ADC-AuNP and siRNA duplex, respectively”.

For the original Figure S8 (i.e., revised Figure 4), in the Supporting information of revised manuscript, we state: “As shown in Fig. 4A, the naked siRNA and siRNA/ADC-AuNP cannot be detected in tumor sites throughout the time course. In contrast, the siRNA/Ap-CS nanoparticles unambiguously accumulated in tumor sites (highlighted with dotted circle) and the fluorescence signal can be detected even at 90 min post-injection, indicating the desirable tumor targeting properties and superior *in vivo* stability. Fig. 4B shows the quantitative contents of siRNAs within tumor sites by fluorescence measurement, implying that the tumor accumulating efficiency of

siRNA/Ap-CS is improved at least by 5.6 times compared with siRNA/ADC-AuNP regardless of incubation time. Fig. 4C demonstrates that siRNA/Ap-CS accumulates in tumor site compared with other organs besides the kidney, and Fig. 4D shows that the content of siRNA/Ap-CS in tumor site is 5.2 times higher than that in liver. Since the kidney serves as an excretory organ through which the nanoparticles are able to be excreted into the urine¹⁷, it is reasonable that a considerable amount of siRNA/Ap-CS is detected in the kidney¹⁷.

Fig. 4. (A) Time-dependent in vivo fluorescence imaging to explore the whole-body biodistribution kinetics of siRNA/Ap-CS and its tumor localization. A549 tumor-bearing BALB/c nude mice were used, and the samples, including PBS, siRNA, siRNA/ADC-AuNP and siRNA/Ap-CS, were administrated to the mice via tail vein injection. (B) The fluorescence intensity of the tumor sites in the white circles in panel A. (C) Fluorescence images of the organs harvested at 90-min post injection of siRNA/Ap-CS, accompanied by the quantitative assessment of fluorescence intensity from liver and tumor (D). The PS-ADC and PA-Cy5 were used for the preparation of siPlk1 duplex.

15) Page 18: The authors need to clarify if they transfected their nanoconstructs into cells, as they state, “including fluorescence image and bright field image by laser scanning confocal microscope, siPlk1/Ap-CS-transfected cells were analyzed by flow cytometry.” I thought these systems did not require transfection vehicles.

Our response: We apologize for our carelessness. The developed nanosystem indeed does not require transfection vehicles. In the revised manuscript, we have changed “siPlk1/Ap-CS-transfected” to “siPlk1/Ap-CS-incubated”.

16) Page 19: The authors need to provide more explanation and clarification as to their conclusion that “for the un-siPlk1/Ap-CS where siRNA cannot be displaced by intracellular miRNA and its fluorescence is partly quenched by AuNP, the treated cells only offer a moderate optical signal (purple line).” It is not intuitive as to why displacement is necessary to measure uptake, and this would answer an earlier question, where an experiment demonstrating quenching upon attachment and release should be provided to the reader.

Our response: We are sorry for our thoughtless description. In fact, besides fluorescence measurement, the cellular uptake of nano-formulations can be evaluated by inductively coupled plasma-mass spectrometry (ICP-MS) as shown in Figure S16. The purpose of this sentence just is to

explain why un-siPlk1/Ap-CS-treated cells display the compromised fluorescence signal compared with siPlk1/Ap-CS-treated cells.

In the revised manuscript, to avoid the misunderstanding, “for the un-siPlk1/Ap-CS where siRNA cannot be displaced by intracellular miRNA and its fluorescence is partly quenched by AuNP, the treated cells only offer a moderate optical signal (purple line) but still lightly higher than that of lipo3000-based transfection” has been changed to “for the un-siPlk1/Ap-CS, the treated cells only show a moderate optical signal (purple line). This is because un-siPlk1 cannot be displaced by intracellular miRNA, and its fluorescence is substantially quenched by AuNP. Nevertheless, the fluorescence intensity is still slightly higher than that of Lipo3000-based transfection”.

Moreover, the fluorescence measurements in the absence and presence of miRNA-21 were carried out to demonstrate the fluorescence quenching of siRNA/Ap-CS unless hybridization-mediated release occurred. As shown in Figure S12C, no obvious fluorescence signal is detected from siPlk1-incorporated formulation unless miRNA-21 is added.

Fig. S12. (C) is the same as (A), but siLuc was substituted with siPlk1. The siPlk1 was obtained by hybridization of Cy5-labeled PA with PS-ADC. After miRNAs (10 μ L, 10 μ M) were separately added into 200 μ L of siPlk1/ADC-AuNP, the fluorescence emission intensity was immediately monitored in real-time.

17) Page 19: There are a few questions regarding the claim that: “Moreover, the internalized siRNA/Ap-CS can successfully escape from endosome/lysosome into the cytosol (SI Appendix, SI Fig. S13), naturally contributing to the efficient gene silencing.” How many cells were imaged for those quantifications?

Our response: The experiments were repeated three times, and at least 30 cells were imaged. The statistical data are represented in Figure S13 mentioned (i.e., Figure S18 in the revised manuscript).

Lysotracker is nonspecific for all acidic compartments so it is hard to say at what point in the endo/lysosomal process these particles left and became non-colocalized. How do their particles compare to other controls, as none were shown in the accompanying data?

Our response: We agree with Reviewer. In this section, we cannot accurately predict the time point of endo/lysosomal process. But, the escape of internalized siRNA/Ap-CS from endosome/lysosome into the cytosol could be estimated by co-localization experiments as shown in Figure S18 where un-siRNA/Ap-CS was used as control. Similar to siRNA/Ap-CS, the internalized un-siRNA/Ap-CS can successfully escape from endosome/lysosome into the cytosol, which is reflected by the

substantial shift of scatter plot. But, no substantial Cy5 fluorescence signal is detected because the siRNA cannot be released by endogenous miRNA, and thus Cy5 fluorescence is always quenched by AuNP.

Fig. S18. The confocal fluorescence images of MCF-7 cells to evaluate the endosome/lysosome escape of siRNA/Ap-CS (A) and un-siRNA/Ap-CS (B) into the cytosol. The Pearson's coefficient (Rr) was calculated by Image J software. Data were presented as mean \pm SD (n = 3).

How can the authors prove that it was not just the fluorophore becoming cleaved from the siRNA and that was escaping the compartments?

Our response: To confirm that the fluorescence signal does not come from the fluorophore cleaved, we monitored the fluorescence emission intensity of Cy5-labeled siPlk1/ADC-AuNP in real-time in 10 % fetal bovine serum (FBS) media according to the method previously reported (J. Am. Chem. Soc. 2018, 140, 1, 258-263). As shown in Figure S12D, compared with the increase in fluorescence intensity upon addition of miRNA-21D, no substantial fluorescence change is detected for another sample, indicating that the fluorescence signal comes from the displacement of fluorescently-labeled siRNA by miRNA-21 rather than the cleavage reaction. Clearly, that Cy5 fluorescence is detected from Figure S18A is because the fluorescently-modified siRNA is indeed released from siRNA/Ap-CS by intracellular miRNA.

Fig. S12. (D) The real-time monitoring of fluorescence emission intensity of siLuc/ADC-AuNP in the presence and absence of miRNA-21.

Where is the time-course of gene knockdown that supports this data?

Our response: We have explored the time-course of gene knockdown, accompanied by the therapeutic efficacy at the cellular level. The measured data are shown in Figure S19. The released siPlk1 does suppress the expression of target gene, resulting in cancer cell death.

Fig. S19. The dependence of gene silencing efficiency on the incubating time for treating the cells with siPlk1/Ap-CS. (A) Plk1 mRNA level in MCF-7 cells treated with siPlk1/Ap-CS for different time periods. (B) The corresponding cell viability. Although Plk1 mRNA level decreases with the incubation time after 15 h, the high therapeutic efficacy is achieved at 48-h incubation. Thus, the 48-h incubation was adopted in the subsequent experiments.

18) Page 19: Why did the authors select 48 h as the timepoint? What about assessing qPCR expression at earlier time points? If they observe “escape” at 10 h, why not try 24h or even 15h? Something closer to 10h?

Our response: In the revised manuscript, we have provided the time-course of gene knockdown efficiency (GKE) and corresponding cell viability (CV). As shown in Figure S19, the experimental results show that, the high therapeutic efficacy is achieved at 48-h incubation, while no substantial change is detected in GKE, especially in CV, at the time points close to 10 h .

19) Many times throughout the paper, the authors say things are “obvious” or “clearly” or “impressive.” These terms should be removed, and the grammar should be corrected to provide more substantiated conclusions based on the data provided.

Our response: We have removed the terms and made the changes in some corresponding sentences. The grammar has also been improved to make the experimental results and conclusions easier to understand.

20) Page 24: Figure 6A/B: Is there significance between the groups? Please provide those stats.

Our response: We have provided the statistical analysis, and the results indicate the significant difference (P<0.01).**

21) SI, Figure S9A: Shouldn't there be two bands in the gel - one for siRNA that was just released and another for the miRNA/ADC complex?

Our response: In this nPAGE image, only the siRNA is modified with fluorophore. Namely, as stated in the figure caption, “FAM-labeled LUC-antisense (LA-FAM) was used for the preparation

of siLuc duplex to monitor fluorescently the hybridization-based strand displacement process". Thus, siRNA or siRNA/ADC complex can be detected, while miRNA/ADC complex is invisible because it does not fluoresce.

22) SI, Figure S11: Where are the appropriate controls that show all images of other conditions?

Our response: In the revised manuscript, we have presented the siLuc/CS, un-siLuc/CS, siLuc/ADC-AuNP and un-siLuc/AD-AuNP as additional controls, and the corresponding fluorescence images are shown in the panels C and D of revised Figure S14.

Fig. S14. Specific membrane receptor-mediated internalization of siRNA/Ap-CS and miRNA-triggered siRNA release, where siLuc is used as the siRNA model. (A) Confocal fluorescence imaging of HeLa cells treated with two different siRNA/Ap-CS formulations. *The upper panel:* releasable siLuc/Ap-CS is used, which is the expected and efficient siRNA-loaded formulation. *The lower panel:* unreleasable un-siLuc/Ap-CS in which LS-AD strand was used to hybridize with LA-FAM to prepare fluorescent siLuc, and AD substituted for ADC strand. (B) The upper and lower panels are the same as the two panels of (A) respectively, but HeLa cells were pre-blocked via incubation with excess AS1411. (C) The upper and lower panels are the same as the two panels of (A) respectively, but siRNA/CS or un-siRNA/CS was instead used. (D) The upper and lower panels are the same as the two panels of (A) respectively, but siRNA/ADC-AuNP or un-siRNA/ADC-AuNP was instead used.

23) SI, Figure S17: For this figure, but also for the flow plots that are shown in general, how many replicates were run? Is this just one experimental batch?

Our response: We repeated the experiments three times using three independent batches. The statistical results have been described in Figure S23 in the revised manuscript.

Fig. S23. Cell viability (A) and apoptosis assay (B) of A549 cells after treatment with different siRNA-incorporated

formulations. Qs 1-4 represent necrotic cells, late apoptotic cells, early apoptotic cells and non-apoptotic cells, respectively. *P<0.05, ***P<0.001, two-tailed unpaired t test. Data were presented as mean \pm SD (n = 3).

Reviewer #2 (Remarks to the Author):

The manuscript entitled “Programmably tiling rigidified-DNA brick on AuNP as multi-functional 3D shell for cancer-targeted delivery of siRNAs” describes the construction of a spherical nucleic acid delivery system with increased serum stability and responsiveness to endogenous stimuli, and demonstrates its application for siRNA delivery in mouse xenograft model. This work addresses important challenges for the systemic application of spherical nucleic acid technology and is a valuable addition to the toolbox of SNA formulation. The authors took a novel approach to create Y-shaped backbone-rigidified triangular DNA bricks that can protect the siRNA from degradation and enable incorporation of the aptamer to enhance targeting of tumor cells. The authors also investigated the use of endogenous miRNA as a trigger to enhance the release of siRNA by a strand displacement reaction. The major criticism is related to the design and conclusions from *in vivo* work and lack of statistical analysis which should be addressed.

Our response: Thanks for your time and valuable comments. As you suggested, new important data (for example, Figures S5, S9, S10, S12, S18B and S19), including the *in vivo* therapeutic efficacy (e.g., Figure S26), have been added, and the appropriate statistical analyses have been represented. The detailed responses are as follow.

Major comments:

1. The authors should include the addition group containing scrambled siRNA to distinguish between the therapeutic efficacy of siRNA/Ap-CS and the effect of tumor accumulation of the targeted particles. siRNA/ADC-AuNP show drastically different accumulation in the tumor than targeted NPs and can't be directly compared. Demonstrating *in vivo* Plk-1 mRNA or protein knockdown would also help strengthen the evidence of therapeutic efficacy.

Our response: We have provided the scrambled siRNA as control according to the reviewer's advices. The *in vivo* tumor accumulation, miRNA knockdown and therapeutic efficacy against malignant tumor in A549 NSCLC xenograft murine model were shown in Figure S26. The scrambled siRNA/Ap-CS can be accumulated at tumor sites but display the negligible treatment efficacy. The expression level of *in vivo* Plk-1 mRNA was explored by qPCR. As shown in Figure S26E, siRNA/Ap-CS can silence the Plk-1 mRNA expression compared with other formulations. Moreover, the comparative study of the whole-body biodistribution of siRNA/ADC-AuNP with siRNA/Ap-CS in tumor-bearing mice is shown in Figure 4 in the revised manuscript. There is a significant difference in tumor accumulation between them.

Fig. S26. *In vivo* tumor accumulation and therapeutic efficacy of siRNA/Ap-CS against malignant tumor in A549 NSCLC xenograft murine model. (A) Time-dependent *in vivo* fluorescence imaging to explore the whole-body biodistribution kinetics of siRNA/Ap-CS and tumor localization. The samples, including scrambled siRNA/Ap-CS, Liposome/siRNA, siRNA/CS and siRNA/Ap-CS, were administrated to the mice via tail vein injection. Cy5-labeled PA and PS-ADC were used for the preparation of siPlk1. (B) The fluorescence intensity of tumor in yellow cycles indicated in (A). (C) Tumor growth curves during treating with siPlk1 delivered by different formulations. The arrows represent the intravenous injections via tail vein. The data are presented as the mean \pm SD of four independent experiments. (D) Images of harvested tumors 14 days after treatment with different siPlk1-encapsulated formulations. (E) Plk1 mRNA levels in harvested tumors were measured by qPCR. ** $P < 0.01$, **** $P < 0.0001$, two-tailed unpaired t test.

2. On page 5 authors claim that their system displays long circulation lifetime. To support this claim authors should provide the pharmacokinetic data such as plasma clearance, the steady-state volume of distribution, and half-life.

Our response: We have explored the pharmacokinetic parameters of siRNA in plasma following tail intravenous injection of siRNA/Ap-CS, siRNA/CS, siRNA/ADC-AuNP, and siRNA duplex in mice. The pharmacokinetic data, including plasma clearance (CL), the steady-state volume of distribution (V_{dss}), and half-life ($t_{1/2}$), are shown in Figure S9.

$$C = C_0 e^{-kt}$$

$$t_{1/2} = \ln 2 / k$$

$$AUC_{0-t} = \int_0^t C dt = C_0 e^{-kt} / -k$$

$$CL = D / AUC$$

$$MRT = \int_0^t tC \frac{dt}{AUC} = \frac{AUMC}{AUC}$$

$$V_{dss} = MRT \times CL$$

	CL (mL/h/kg)	V_{dss} (mL/kg)	$t_{1/2}$ (h)
siRNA/Ap-CS	8.81 ± 0.32	849.10 ± 0.26	1.34 ± 0.05

siRNA/CS	10.89±0.84	751.60±47.91	0.96±0.04
siRNA/ADC-AuNP	17.37±0.26	653.89±12.77	0.52±0.01
siRNA	59.73±15.02	775.97±51.86	0.19±0.05

Fig. S9. The pharmacokinetic profiles of siRNA-incorporated formulations in mice. Cy5-labeled PA and PS-ADC were used for the preparation of siPlk1. The pharmacokinetic parameters in the equations, C , C_0 , k , $t_{1/2}$, AUC, CL, D, MRT, and V_{dss} , represent plasma siRNA concentration, initial plasma siRNA concentration, elimination constant, time, plasma half-life, area under the curve, clearance, dose, mean residence time in serum, and distribution volume at steady state, respectively.

Based on the data presented in Figure S8 it is not clear whether including the outer coating can influence the circulation times *in vivo*, which could also contribute to tumor delivery. This is important in light of the data from Figure 5, showing that enhanced stability provided by the coating does not significantly contribute to the silencing efficacy in the xenograft model.

Our response: To address the issues raised, we have explored the pharmacokinetic parameters of the different siRNA-incorporated formulations (seen in Figure S9) and evaluated the *in vivo* tumor accumulation and therapeutic efficacy (Figure S26). The experimental results demonstrate that, although the outer coating increases the circulation time of siRNA/CS, the silencing efficacy and therapeutic efficacy are not satisfactory, which is consistent with the results in the revised Figure 6 (Figure 5 mentioned by Reviewer). This is because of the lack of aptamer-based tumor targeting, resulting the undesirable tumor accumulation. In contrast, on the basis of the combination of protective outer coating and aptamers, siRNA/Ap-CS shows the high tumor accumulation and desirable therapeutic efficacy.

3. The targeted siRNA/Ap-CS particles show 6-fold higher accumulation in the liver as compared with siRNA/ADC-AuNP particles (Figure S) but the authors do not provide any explanation for this observation in the manuscript and the extended discussion S3.

Our response: The higher accumulation in the liver is owing to the increased resistance to the nuclease degradation, which has been reported by William M. Shih and colleagues (Nat. Commun. 2017, 8, 15654 that is cited in Discussion S3). The corresponding explanation has been provided in discussion S3: "These experimental results are consistent with the previous observations¹⁶, demonstrating that siRNAs loaded in the core/shell nanostructure are protected from the nuclease degradation in systemic circulation and exhibit the enhanced stability".

4. Given the strong accumulation of the siRNA/Ap-CS particles in the liver, the manuscript would greatly benefit from including the liver toxicity data (e.g. ALS, AST, AP) to support the claims on the systemic toxicity, which authors based solely on observing the changes in animal weight and behavior after treatment.

Our response: We have provided the liver toxicity data (ALS, AST and AP). As shown in Figure S10, the experimental results show that siRNA/Ap-CS treatment does not cause significant changes in AST, ALT and AP levels, compared with saline controls.

Fig. S10. The liver toxicity of mice intravenously injected with siRNA/Ap-CS or PBS for 48 h. The alanine aminotransferase (ALT), aspartate aminotransferase (AST) and alkaline phosphatase (AP) were measured by ALT kit, AST kit, and AP kit, respectively.

5. It is not clear how the siRNA encapsulation efficacy and the final dose were calculated for the developed system used for *in vivo* studies.

Our response: The numbers of siRNA and ADC in siRNA/Ap-CS are 420 and 720 per AuNP, respectively (seen in the section of Methods, “Construction of siRNA/Ap-CS” and “Quantification of anchoring DNAs covalently attached to AuNPs”). Theoretically, 720 siRNA can be loaded per AuNP. Hence, the siRNA encapsulation efficacy is 58.3 % (estimated from $420/720 \times 100\%$).

The final dose for *in vivo* studies is 100 μL (the siRNA concentration, 0.91 μM) and nano-formulation was intravenously injected via tail vein every two days (seen in the section of Methods, “Gene silencing performance of siRNA/Ap-CS formulation *in vivo*”).

To also allow a better comparison with other delivery modalities, the siRNA dose used for *in vivo* studies should be expressed in mg/kg.

Our response: Taking into account that the final dose of siRNA for *in vivo* studies was 100 μL (0.91 μM), the molecular weight of Plk1 siRNA is 20397.37 $\mu\text{g}/\mu\text{mole}$ and average mouse weight is 20 g, the siRNA dose used for *in vivo* studies is approximately 0.1 mg/kg [estimated from $(0.91 \mu\text{M} \times 20397.37 \mu\text{g}/\mu\text{mole} \times 100 \mu\text{L})/20 \text{ g}$].

6. Authors should perform appropriate statistical analysis and indicate the statistically significant difference for they're *in vivo* and *in vitro* data (Fig 5, 6, S12, S16, S17)

Our response: We have offered the appropriate statistical analyses in Figures 5, 6, S12, S16 and S17 (namely, Figures 6, 7, S15, S23 and S24 in the revised manuscript).

Minor comments:

1. On page 15: Authors cite the work of Jeong et al. Bioconjugate Chem. 20, 5-14 (2009) and state that “chemical modification of siRNAs compromises the silencing activity” in the context of using chemical modifications of siRNA with AuNP. This statement is misleading and should be withdrawn or corrected since neither the authors nor the cited work has shown the validity of this with AuNP and this effect will likely depend on the type of modifications used. In fact, the majority of clinically evaluated siRNAs and approved drugs like Parisian and Givosiran use modified siRNAs.

Our response: We apologize for the misleading statement. In the revised manuscript, the “chemical modification of siRNAs compromises the silencing activity” has been changed to “chemical

modification of siRNAs, such as at the 5'-terminus of antisense strand, potentially compromises the silencing activity".

REVIEWER COMMENTS

Reviewer #1 (Remarks to the Author):

In this work, the authors developed an siRNA delivery system, where siRNA is hybridized to a complementary thiol-conjugated DNA strand covalently attached to gold nanoparticles and covered with aptamer-incorporated, Y-shaped backbone-rigidified triangular DNA bricks. The authors demonstrate the ability of these nanoparticles to target cancer cells expressing the aptamer binding domain and appropriately release siRNA through binding and displacement. Once the siRNA has been released from the nanoparticle, they demonstrate in vitro gene knockdown through mRNA expression and western blots of protein levels of PLK1, a regulator of mitotic progression in mammalian cells that is overexpressed in cancer cells. In addition, they apply this system to an in vivo xenograft model, where they demonstrate after 14 days a ~3-fold reduction in tumor volume compared to siRNA treatment alone.

While the authors did address many of my points, unaddressed issues remain – these should be resolved prior to the publication of this manuscript:

1) Page 11: The authors show modifications in the legend to the gels in Figure 2C that highlight the electrophoretic mobility of the different constructs. However, they do not address why there is a faint blue band for each well. In addition, the red bands are very faint especially the first and last lanes, thus it's hard for the reader to see them.

2) Page 12: The authors say that the red shift of the UV-vis spectrum in Figure 2E is indicative of the successful assembly of the nanoparticles, and they provide references in their reply. This is fine, but zeta potential would be a much easier way to visualize this data and would better show the differences between the constructs. In addition, the authors can provide zeta potential measurements of each step of the assembly, as was done in Fig. 2C.

3) Page 13: Remove "apparently" and change it to "We believe that." The authors clarify in the reply that the FBS stability of siRNA/Ap-CS is higher than those under the double protection reported in a previous study (Proc. Natl. Acad. Sci. U.S.A. 2014, 111, 9739-9744). However, this comparison still doesn't appear logical. They are specifically crosslinking their structural units together in addition to providing a dense shell of blocking units. Thus, the chemical modification of a 2'-OMe just isn't a fair comparison. A cross-linked structure without their YTDDBs would be better as a comparison based on the wording in the text. The extra experiment in S5 is fine and does articulate half of my point as it clarifies one comparison. But, it does not compare crosslinking without the triangular brick unit.

4) Page 16, Fig 4: The authors should also include the data showing the aptamer (AS1411) installed onto the outermost protective layer of 3D DNA shell - in particular, Figures S13 and S14 - if the data can be quantitatively presented.

5) Page 15: Regarding the claim that the "stimulus-responsive release of siRNA can be realized because the ADC is capable of preferentially hybridizing with endogenous miRNA-21": In the Figure S12 provided, the authors should show the turn on in fluorescence. The signal of miRNA-21, while higher than the others, never appears to go through a turn-on event. It would be nice to see this change to definitively validate this claim, as shown in S12B and C. The data provided in S12B and C is very valuable.

6) Page 16: The authors do not address my issue. I understand that the 73% calculated for silencing was based on the assumption that the negative control was 100% expression, but they do not articulate how that number was obtained. From my understanding, it was through luminescence via a plate reader. The mention of determination of gene knockdown through luminescence measurement needs to be in the main text.

7) Page 16: The reply and revision regarding the claim that "a strong red fluorescence signal is detected in fluorescence image (Cy5) and a considerable number of AuNPs, [...] demonstrating the successful cellular uptake of siPlk1/Ap-CS and intracellular miRNA-triggered the efficient release of formulated siPlk1" is much better restated and expanded. However, please increase the brightness of Figures 5A and B in the red Cy5 channel so that the color can be better observed. The reply to the additional point about quenching of the Cy5 I believe should reference Figure S12D, not S12C.

8) Page 17: In Figure 4 C, please label each of the gray boxes for what each sample/mouse was.

9) Page 19: The rewording of the text to "unsatisfactory resistance to nuclease-induced

degradation in serum" is confusing. That sentence instead could just be stated: But while significantly superior to their linear counterparts, SNAs still suffer from two inherent limitations: the extensive long-term in vivo susceptibility to nuclease degradation and the lack of targeting specificity 27, 32."

10) Page 17: The authors state that their system has overcome the bottleneck of in vivo delivery to the tumor site. While their reply to this comment provided much information, ultimately what would be most helpful to the reader is a comparison of the tumor:liver ratio for all three samples. While the increased tumor accumulation is noted quite well in Figure 4, Figure S8 confuses me because of the increased liver accumulation. These are at very similar time points (60 vs. 90 min), and thus it is very confusing to understand why the data in the SI doesn't appear to match the images in Figure 4C, where the liver accumulation looks low. Thus, a simple way for the reader to best understand this point would be to just provide a comparison of the tumor:liver ratio for all three samples.

11) Page 19: The response to the questions about the endo/lysosome escape are mostly addressed except it would still be nice to have a control that is incapable of escaping the compartments.

12) In the SI, Figure S23: Why doesn't the data provided in the flow plot match the bar graph of viability? I understand that there is some error and that the flow is a representative sample, but for siRNA/Ap-CS, for example, the bar in A is around 70% and the flow in Q4 is 45%. The error bar of SD doesn't look big enough to include such a sample.

Reviewer #2 (Remarks to the Author):

The authors adequately addressed my comments and included additional data that significantly strengthened the quality of this impressive work.

REVIEWER COMMENTS

Reviewer #1 (Remarks to the Author):

In this work, the authors developed an siRNA delivery system, where siRNA is hybridized to a complementary thiol-conjugated DNA strand covalently attached to gold nanoparticles and covered with aptamer-incorporated, Y-shaped backbone-rigidified triangular DNA bricks. The authors demonstrate the ability of these nanoparticles to target cancer cells expressing the aptamer binding domain and appropriately release siRNA through binding and displacement. Once the siRNA has been released from the nanoparticle, they demonstrate in vitro gene knockdown through mRNA expression and western blots of protein levels of PLK1, a regulator of mitotic progression in mammalian cells that is overexpressed in cancer cells. In addition, they apply this system to an in vivo xenograft model, where they demonstrate after 14 days a ~3-fold reduction in tumor volume compared to siRNA treatment alone.

While the authors did address many of my points, unaddressed issues remain – these should be resolved prior to the publication of this manuscript:

1) Page 11: The authors show modifications in the legend to the gels in Figure 2C that highlight the electrophoretic mobility of the different constructs. However, they do not address why there is a faint blue band for each well. In addition, the red bands are very faint especially the first and last lanes, thus it's hard for the reader to see them.

Our response: Thanks for your constructive comments. The faint blue band for each well should be attributed to the fluorescence of Bromophenol Blue existing in commercial loading buffer that possibly also influences the quality of product bands. To obtain the clear bands, we used glycerin: EDTA (50 mM) at the volume ratio of 4:1 instead of loading buffer. Moreover, all the samples were freshly prepared for performing the agarose gel analysis. Specifically, each sample was prepared by mixing 10 μ L of naked AuNP or nucleic acid-modified AuNP with 5 μ L of mixture of glycerin and EDTA (50 mM) at the volume ratio of 4:1. The new Fig. 2C is shown below.

2) Page 12: The authors say that the red shift of the UV-vis spectrum in Figure 2E is indicative of the successful assembly of the nanoparticles, and they provide references in their reply. This is fine, but zeta potential would be a much easier way to visualize this data and would better show the differences between the constructs. In addition, the

authors can provide zeta potential measurements of each step of the assembly, as was done in Fig. 2C.

Our response: The zeta potential analysis at different stages during the assembly of siRNA/Ap-CS have been performed and the experimental results are displayed as Fig. 2D. The corresponding statement has been added: “Fig. 2D shows that zeta potential of four nanostructures sequentially decreases, indicating the increasing amount of nucleic acids assembled onto AuNP surface”.

Fig. 2. (D) Zeta potential of four nanostructures in C determined by DLS.

3) Page 13: Remove “apparently” and change it to “We believe that.”

Our response: We have changed “apparently” to “We believe that” in the revised manuscript.

The authors clarify in the reply that the FBS stability of siRNA/Ap-CS is higher than those under the double protection reported in a previous study (Proc. Natl. Acad. Sci. U.S.A. 2014, 111, 9739-9744). However, this comparison still doesn't appear logical. They are specifically crosslinking their structural units together in addition to providing a dense shell of blocking units. Thus, the chemical modification of a 2'-OMe just isn't a fair comparison. A cross-linked structure without their YTDBs would be better as a comparison based on the wording in the text. The extra experiment in S5 is fine and does articulate half of my point as it clarifies one comparison. But, it does not compare crosslinking without the triangular brick unit.

Our response: To construct a cross-linked core/shell nanostructure (cross-linked CS) without YTDBs, S1' was used for the assembly instead of the hybrid consisting of S1, S2 and S3, while other constituents, including the assembly procedure, are the same as siRNA/Ap-CS. Thus, the surface of assembled products has the palindromic end-based intermolecular linking effect but without YTDBs. Its FBS stability was explored under identical conditions, and the results are shown in the revised Fig. S5. One can notice that the residual amount of siRNAs encapsulated in the four formulations decreases in the order of Nanoparticle-a > Nanoparticle-b > Nanoparticle-c > Nanoparticle-d.

Fig. S5. Comparative study of the FBS stability of siRNA/Ap-CS with the counterpart nano-formulation under double protection by SNA and 2'-OMe or by cross-linked core/shell nanostructure (cross-linked CS). Quantitative evaluation of the retention (%) of surface-confined siRNAs was performed after exposure of different siRNA-encapsulated formulations (a-d) to FBS, where the retention efficiency of nontreated formulation corresponding to each group is defined as 100%. Nanoparticle-a is the siRNA/Ap-CS with the protective outer coating, Nanoparticle-b is the cross-linked CS, Nanoparticle-c is 2'-OMe-siRNA/ADC-AuNP with the double protective effect of SNA structure and 2'-OMe modification, and Nanoparticle-d is siRNA/ADC-AuNP only with the protective SNA structure.

4) Page 16, Fig 4: The authors should also include the data showing the aptamer (AS1411) installed onto the outermost protective layer of 3D DNA shell - in particular, Figures S13 and S14 - if the data can be quantitatively presented.

Our response: The data mentioned by Reviewer has been presented in the supporting information. Specifically, the comparative evaluation between the formulations with and without aptamers is described in Fig. S26. The aptamer-anchored one (siRNA/Ap-CS) displays the significantly higher capability to accumulate in tumor tissues than siRNA/CS without aptamer (Fig. S26A and S26B), leading to the targeted therapy efficacy confirmed by the substantial increase in tumor volume (Fig. S26C and S26D) and considerable expression of Plk1 mRNA expression (Fig. S26E). Moreover, siRNA alone and siRNA/ADC-AuNP formulation have also been evaluated as the other two controls (Fig. 4). More controls are seen in Figure S14.

In addition, the quantitative value of the fluorescence intensity of each samples in Fig. S13 and Fig. S14 was measured by Image J software, and the data were presented as mean \pm SD (n = 3) in the revised Supporting Information.

Fig. S26. *In vivo* tumor accumulation and therapeutic efficacy of siRNA/Ap-CS against malignant tumor in A549 NSCLC xenograft murine model. (A) Time-dependent *in vivo* fluorescence imaging to explore the whole-body biodistribution kinetics of siRNA/Ap-CS and tumor localization. The samples, including scrambled siRNA/Ap-CS, Liposome/siRNA, siRNA/CS and siRNA/Ap-CS, were administrated to the mice via tail vein injection. Cy5-labeled PA and PS-ADC were used for the preparation of siPlk1. (B) The fluorescence intensity of tumors in yellow cycles indicated in (A). (C) Tumor growth curves during treating tumor with siPlk1 delivered by different formulations. The arrows represent the intravenous injections via tail vein. The data are presented as the mean \pm SD of four independent experiments. (D) Images of harvested tumors 14 days after treatment with different siPlk1-encapsulated formulations. (E) Plk1 mRNA levels in harvested tumors were measured by qPCR. ** $P < 0.01$, **** $P < 0.0001$, two-tailed unpaired t test.

5) Page 15: Regarding the claim that the “stimulus-responsive release of siRNA can be realized because the ADC is capable of preferentially hybridizing with endogenous miRNA-21”: In the Figure S12 provided, the authors should show the turn on in fluorescence. The signal of miRNA-21, while higher than the others, never appears to go through a turn-on event. It would be nice to see this change to definitively validate this claim, as shown in S12B and C. The data provided in S12B and C is very valuable.

Our response: To act on the advice, we have revised the Fig. S12B and C. To perform a turn-on fluorescence change, the samples were first scanned for 200 s, and then the miRNA were added and scanned for another 1000 s. Just as we designed, the fluorescence is weak 200 s before addition of miRNA-21, while the fluorescence rapidly increases upon the miRNA-21 working as a stimulus.

Fig. S12. (A) The real-time monitoring of fluorescence emission intensity of siLuc/ADC-AuNP in the presence of different miRNAs (miRNA-21, miRNA-429, miRNA-141, and miRNA-200b). The siLuc duplex was prepared by hybridizing LS-ADC with FAM-labeled LUC-antisense (LA-FAM). (B) and (C) are the same as (A), but siLuc was substituted with dsDNA and siPlk1, respectively. The dsDNA was prepared by hybridizing FAM-labeled ES-ADC to EAH, while siPlk1 was obtained by hybridization of Cy5-labeled PA with PS-ADC. The red arrow at 200 s represents the time point of miRNA stimulus. (D) The real-time monitoring of fluorescence emission intensity of siPlk1/ADC-AuNP in the presence and absence of miRNA-21. (E) The real-time monitoring of

fluorescence emission intensity of un-siPlk1/AD-AuNP in the presence and absence of miRNA-21, where un-siPlk1 was prepared by hybridizing PS-AD with PA-Cy5.

6) Page 16: The authors do not address my issue. I understand that the 73% calculated for silencing was based on the assumption that the negative control was 100% expression, but they do not articulate how that number was obtained. From my understanding, it was through luminescence via a plate reader. The mention of determination of gene knockdown through luminescence measurement needs to be in the main text.

Our response: The evaluation of gene knockdown efficiency was measured by a luminometer (Tecan Infinite 200 pro, Switzerland), and this formation has been added into the revised manuscript. We have stated: “The expression level of luciferase was determined by luminescence measurement in luminometer (Tecan Infinite 200 pro, Switzerland), and if the expression level of luciferase in HeLa cells treated with PBS is defined as 100%, only 27% luciferase is expressed after being treated with siRNA/Ap-CS. Namely, about 73% target genes is silenced, which is comparable with the commercial transfection reagent.”

7) Page 16: The reply and revision regarding the claim that “a strong red fluorescence signal is detected in fluorescence image (Cy5) and a considerable number of AuNPs, [...] demonstrating the successful cellular uptake of siPlk1/Ap-CS and intracellular miRNA-triggered the efficient release of formulated siPlk1” is much better restated and expanded. However, please increase the brightness of Figures 5A and B in the red Cy5 channel so that the color can be better observed.

Our response: Thanks. The fluorescence in Fig. 5A and Fig. 5B has been increased.

The reply to the additional point about quenching of the Cy5 I believe should reference Figure S12D, not S12C.

Our response: We have changed “The specific release of Cy5-labelled siPlk1 from the nano-formulation is verified by fluorescence measurement as shown in Fig. S12C” to “The specific release of Cy5-labelled siPlk1 from the nano-formulation is verified by fluorescence measurement as shown in Fig. S12D”.

8) Page 17: In Figure 4 C, please label each of the gray boxes for what each sample/mouse was.

Our response: We have added the corresponding label for each of samples in the bottom in Fig. 4C.

C

Fig. 4 (C) Fluorescence images of the organs harvested at 90-min post injection of siRNA/Ap-CS.

9) Page 19: The rewording of the text to “unsatisfactory resistance to nuclease-induced degradation in serum” is confusing. That sentence instead could just be stated: But while significantly superior to their linear counterparts, SNAs still suffer from two inherent limitations: the extensive long-term in vivo susceptibility to nuclease degradation and the lack of targeting specificity 27, 32.”

Our response: Thanks for your insightful advice. In the revised manuscript, we have changed “But it suffers from two inherent limitations: the unsatisfactory resistance to nuclease-induced degradation in serum and the lack of targeting specificity^{27, 32}” to “But while significantly superior to their linear counterparts, SNAs still suffer from two inherent limitations: the extensive long-term in vivo susceptibility to nuclease degradation and the lack of targeting specificity^{27, 32}”.

10) Page 17: The authors state that their system has overcome the bottleneck of in vivo delivery to the tumor site. While their reply to this comment provided much information, ultimately what would be most helpful to the reader is a comparison of the tumor: liver ratio for all three samples.

Our response: We agree with the Reviewer. We have added the fluorescence ratio of tumor-to-liver in Fig. 4D. One can easily notice that siRNA/Ap-CS does preferentially accumulate in tumor tissue.

Fig. 4. (C) Fluorescence images of the organs harvested at 90-min post injection of siRNA/Ap-CS, accompanied by the quantitative assessment of fluorescence intensity from liver and tumor (D) where the fluorescence ratio of tumor-to-liver (T/L) is also presented.

While the increased tumor accumulation is noted quite well in Figure 4, Figure S8 confuses me because of the increased liver accumulation. These are at very similar time points (60 vs. 90 min), and thus it is very confusing to understand why the data in the SI doesn't appear to match the images in Figure 4C, where the liver accumulation looks low. Thus, a simple way for the reader to best understand this point would be to just provide a comparison of the tumor: liver ratio for all three samples.

Our response: The obvious difference in fluorescence intensity between 60-min and 90-min post injection of siRNA-incorporated formulations is described in Fig. 4B. Moreover, it is reasonable that the fluorescence intensity from liver presented in Fig. 4 is different from that in Fig. S8. This is because the tumor-bearing mice were used in the former, while normal mice were involved in the latter. The experimental results (siRNA/Ap-CS reaching liver organ) shown in Fig. S8 is consistent with the literature report (Nat. Commun. 2017, 8, 15654), indicating the enhanced in vivo stability of siRNA/Ap-CS, which has been discussed in our manuscript. For Fig. S8A describes the biodistribution of several siRNA formulations at 1 h following post-injection into normal mice, and each group of mice show the fluorescence signal that varies in the intensity in different regions. For the siRNA/Ap-CS group, the fluorescence signal mainly focuses on the midsection. To offer accurate evaluation, the mice were killed and their organs were harvested for fluorescence imaging. As shown in Fig. S8B, all the siRNA formulations preferentially distribute in the liver rather than in other organs, but there is a significant difference in the fluorescence intensity from the liver between the three siRNA formulations. Fig. S8C shows the quantitative measurements of siRNA formulations distributed in the livers, confirming the substantial accumulation of siRNA/Ap-CS in the liver compared with naked siRNA and siRNA/ADC-AuNP. These experimental results are consistent with the previous observations¹⁶, demonstrating that siRNAs loaded in the core/shell nanostructure are protected from the nuclease degradation in systemic circulation and exhibit the enhanced stability". Nevertheless, to act on the advice, the comparison of fluorescence ratio of tumor-to-liver ratio between all three samples has been provided in Fig. 4D.

11) Page 19: The response to the questions about the endo/lysosome escape are mostly addressed except it would still be nice to have a control that is incapable of escaping the compartments.

Our response: To construct the corresponding structure incapable of escaping the endo/lysosome, siRNA/AP-YTDB without AuNP was used as a control, and the experimental results are shown in Fig. S18C. One can notice that siRNA/AP-YTDB cannot escape from the endo/lysosome. Therefore, the corresponding statement has been added: “For siRNA/AP-YTDB used as another control, the change of Pearson’s coefficient shows a different trend (Scatter plot) (0.07 ± 0.01 at 2-h incubation, 0.12 ± 0.02 at 4-h incubation and 0.15 ± 0.12 at 10-h incubation), demonstrating the entrapment of siRNA/AP-YTDB in endosome/lysosome. Moreover, the obvious Cy5 fluorescence signal is detected, indicating the separation of Cy5 from the quencher owing to the degradation of nucleic acids”.

Fig. S18. The confocal fluorescence images of MCF-7 cells to evaluate the endosome/lysosome escape of siRNA/AP-CS (A), un-siRNA/AP-CS (B) and siRNA/AP-YTDB (C) into the cytosol. The Pearson’s coefficient (R_r) was calculated by Image J software. Data were presented as mean \pm SD ($n = 3$).

12) In the SI, Figure S23: Why doesn’t the data provided in the flow plot match the bar graph of viability? I understand that there is some error and that the flow is a representative sample, but for siRNA/AP-CS, for example, the bar in A is around 70% and the flow in Q4 is 45%. The error bar of SD doesn’t look big enough to include such a sample.

Our response: We have performed flow cytometry of each sample, including siRNA/AP-CS, five more times, and the new data are shown in the revised Fig. S23. There still is small difference between panel B and panel B, which is similar to the literature reports (ACS Nano 2019, 13, 260–

273; ACS Appl. Mater. Interfaces 2020, 12, 56782–56791). This should be attributed to the difference between flow cytometry analysis and CCK8-kit assay.

Fig. S23. Cell viability (A) and apoptosis assay (B) of A549 cells after treatment with different siRNA-incorporated formulations. Qs 1-4 represent necrotic cells, late apoptotic cells, early apoptotic cells and non-apoptotic cells, respectively. * $P < 0.05$, *** $P < 0.001$, two-tailed unpaired t test. Data were presented as mean \pm SD ($n > 3$).

Reviewer #2 (Remarks to the Author):

The authors adequately addressed my comments and included additional data that significantly strengthened the quality of this impressive work.

Our response: Thanks for our efforts and support.

REVIEWERS' COMMENTS

Reviewer #1 (Remarks to the Author):

The authors have addressed all of my comments, and thus I recommend this manuscript for publication without further changes.